# HBLLM: Wavelet-Enhanced High-Fidelity 1-Bit Quantization for LLMs

**Ningning Chen**[1,*]
chennn27@mail2.sysu.edu.cn

**Weicai Ye**[1,2*]
cai_rcy@163.com

**Ying Jiang**[1,2†]
jiangy32@mail.sysu.edu.cn

[1]Sun Yat-sen University
[2]Guangdong Province Key Laboratory of Computational Science

## Abstract

We introduce HBLLM, a wavelet-enhanced high-fidelity 1-bit post-training quantization method for Large Language Models (LLMs). By leveraging Haar wavelet transforms to enhance expressive capacity through frequency decomposition, HBLLM significantly improves quantization fidelity while maintaining minimal overhead. This approach features two innovative structure-aware grouping strategies: (1) frequency-aware multi-parameter intra-row grouping and (2) $\ell_2$-norm-based saliency-driven column selection. For non-salient weights, a shared mean is employed across quantization groups within each frequency band to optimize storage efficiency. Experiments conducted on the OPT and LLaMA models demonstrate that HBLLM achieves state-of-the-art performance in 1-bit quantization, attaining a perplexity of 6.71 on LLaMA2-13B with an average weight storage of only 1.08 bits. Code available at: https://github.com/Yeyke/HBLLM.

## 1 Introduction

In recent years, Large Language Models (LLMs) have achieved remarkable progress in natural language processing tasks. However, their massive parameter sizes—often reaching tens or even hundreds of billions—pose significant challenges for deployment on edge devices and in low-resource environments. To reduce the computational and memory burden of these models, a variety of compression techniques have been proposed, including quantization [12, 33, 35], pruning [11, 31], and knowledge distillation [19, 30]. Among them, Post-Training Quantization (PTQ) is widely adopted for its efficiency, requiring no additional training and having low deployment cost, especially in 1-bit quantization, which is considered a key approach for achieving extreme inference efficiency [13].

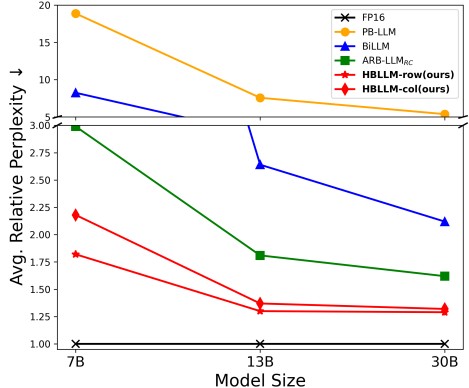

Figure 1: Average relative perplexity (normalized to FP16) on PTB, WikiText2, and C4 for LLaMA-1 family models, comparing LLM binarization methods and our HBLLM.

Although existing 1-bit PTQ methods [15, 17, 34] have achieved some success on base models such as GPT-2 and OPT, they tend to suffer from significant performance degradation—or even complete failure—when applied to more complex modern architectures like LLaMA3-8B [16]. To address this, recent studies have introduced several strategies to improve quantization fidelity:

---

*Equal contribution.
†Corresponding Author.

39th Conference on Neural Information Processing Systems (NeurIPS 2025).

- **Group quantization**: divides the weight matrix into multiple groups for separate quantization. For instance, outlier-aware partitioning handles critical columns independently but can be constrained by partition design and scalability [15];

- **Residual approximation**: adds residual terms on top of primary quantization to partially recover errors [6], though this provides limited fidelity gains and introduces extra computation;

- **Low-Rank Adaptation (e.g., LoRA)**: inserts low-rank modules to absorb quantization errors with some flexibility, like [34], but often shows sensitivity to hyperparameters;

- **Global orthogonal transformations**: apply global rotations in [1, 2, 5] before model compression to enhance representational capacity, but require expensive inverse transforms (e.g., matrix multiplications at $\mathcal{O}(d^2)$ complexity for a $d$-dimension linear layer), leading to increased inference latency and energy consumption, making them impractical for deployment.

To overcome the structural trade-off between expressiveness and efficiency, we propose a novel 1-bit PTQ framework—**HBLLM**. This method is the first to integrate localized orthogonal transformations (i.e., Haar wavelets) into a BiLLM-style quantization process. Combined with structure-aware grouping, HBLLM significantly enhances expressive power under ultra-low bit budgets while maintaining negligible inverse transform cost and excellent compatibility with hardware-efficient inference.

Our main contributions are as follows:

- A **localized orthogonal transformation mechanism**: we apply a single Haar wavelet transform to decompose the weight matrix into high- and low-frequency components, improving binary expressiveness while reducing transform computation;

- **Frequency-aware multi-parameter intra-row grouping**: we introduce intra-row grouping in the frequency domain to capture structural patterns;

- $\ell_2$**-norm-based saliency-driven column selection**: we propose an $\ell_2$ norm-based ranking method to retain key columns using saliency metrics, effectively reducing quantization error;

- **Intra-frequency-band mean sharing**: for non-salient components, we introduce a mechanism that shares the mean across groups within the same row and wavelet band, reducing storage without sacrificing fidelity.

We conduct extensive experiments on OPT [37], LLaMA family [32] of LLMs. Results show that HBLLM achieves state-of-the-art performance under 1-bit quantization: Across language modeling tasks (C4, PTB, WikiText2), the perplexity ratio between HBLLM and the original FP16 model remains within the range of 1.2–2.2, shown in Fig 1, outperforming the next-best methods by 33%–66%; On 9 zero-shot QA benchmarks, HBLLM retains 73.8%–88.8% of the original model's accuracy; On modern architectures such as LLaMA3-8B, HBLLM remains stable with no performance collapse; Even with a lower average bit rate and memory usage than BiLLM and ARB-LLM$_{RC}$ [18], HBLLM outperforms both in overall task accuracy.

These results demonstrate that HBLLM significantly extends the applicability of 1-bit quantization, balancing extreme compression with high fidelity, and offers a new paradigm for deploying large-scale language models efficiently.

## 2 Related Work

### 2.1 1-Bit Post-Training Quantization

1-bit PTQ has emerged as a critical promising solution for deploying LLMs under extremely low bit budgets. Representative methods such as BiLLM [15] adopt a salient column separation mechanism, in which salient weights are quantized independently, while non-salient weights are grouped based on magnitude and quantized row-wise. ARB-LLM$_X$ [18] further introduces column-wise grouping and alternating refined binarization, achieving notable improvements in fidelity. Unlike [10, 34], BiLLM can accomplish PTQ tasks without intensive computation for knowledge distillation with multi-GPUs.

However, current methods face several key limitations: (1) They heavily rely on fixed thresholds or simple $\ell_1$-based heuristics for salient column selection, which are insufficient to capture sparse but significant activation outliers; (2) They fail to account for the structural asymmetry between row and

column dimensions in weight matrices, limiting their adaptability to complex model architectures; (3) They completely neglect frequency-domain information.

## 2.2 Evolution and Limitations of Grouping Strategies

To improve quantization flexibility and fidelity, some studies have proposed learnable or adaptive grouping strategies. For example, Mixture of Scales [17] introduces a Mixture-of-Experts (MoE) mechanism to assign scaling factor groups, and OneBitGPT [34] uses frequency masks to control quantization range sensitivity, and AWQ [3] identifies weights with the greatest impact on model predictions only based on activation outputs. However, these methods are generally effective only on unstructured tensors, rely on fine-grained distillation, and lack explicit frequency-domain awareness.

In addition, existing grouping strategies [15, 18] often apply uniform partitioning rules across the entire weight matrix, ignoring variations across different rows. This can lead to degraded expressiveness when quantizing models with significant inter-row diversity.

## 2.3 Comparison Between Global Orthogonal Transforms and Local Wavelet Transforms

Orthogonal transforms have recently been adopted to improve LLM quantization. FrameQuant [1] and QuIP [5] utilize orthogonal transforms to enhance fidelity, but inference with such global transforms incurs high overhead, requiring $\mathcal{O}(d^2)$ matrix multiplications [1] that cannot be fused into linear layers, leading to increased latency and energy cost.

By contrast, local orthogonal transforms such as the Haar wavelet [20] offer localized spectral sensitivity and have been widely applied in image compression, denoising, and edge detection [9, 14]. They can be efficiently implemented via lightweight local convolutions with negligible inference cost, making them well-suited for low-bit compression and edge deployment.

# 3 HBLLM: A Quantization Framework with Wavelet Transform and Frequency-Domain Grouping

## 3.1 Motivation and Core Challenges

Current mainstream 1-bit quantization methods face three key challenges in practice: (1) limited numerical expressiveness leading to high reconstruction error; (2) insufficient accuracy in salient column selection, failing to capture critical activation columns; (3) lack of structure-aware grouping strategies that adapt to heterogeneous model structures.

To characterize expressiveness under ultra-low bit settings, we introduce a new metric: the *cardinality of the Inverse Quantization Set (CIQ)*, which measures the size of the discrete set of dequantized values within a row. CIQ serves as a unified indicator of how the above challenges constrain model fidelity. It acts both as a theoretical tool to analyze the limits of existing methods and as empirical evidence of the advantage of our proposed method.

Under 1-bit quantization, the CIQ of BiLLM and ARB-LLM$_X$ is 8 and 10, respectively. When block size sets to 128, the CIQ upper bound of ARB-LLM$_X$ can reach 128. In contrast, our method achieves a CIQ of up to 1024 after applying the Haar wavelet transform, significantly improving theoretical expressiveness. For more information on the benefits introduced by applying Haar transform, please refer to the appendix B and C.

Based on aboved analysis, we propose: (1) Haar wavelet transform to enhance expressive capacity by frequency decomposition; (2) $\ell_2$-norm-based saliency-driven column selection to prioritize critical columns; (3) frequency-aware multi-parameter intra-row grouping to capture structural patterns. We also introduce an intra-frequency-band mean sharing strategy and local convolution optimization to reduce storage and inference cost, thus forming a 1-bit PTQ framework **HBLLM** .

## 3.2 Method Overview

We define the objective of HBLLM under the binary quantization setting for LLM weights. Specifically, the quantization targets the full-precision weight matrix $\mathbf{W}_{\text{FP}} \in \mathbb{R}^{d \times d}$, where a binary diagonal mask matrix $\mathbf{M}_{\text{sal}} \in \{0, 1\}^{d \times d}$ indicates which columns are selected as salient. The salient and

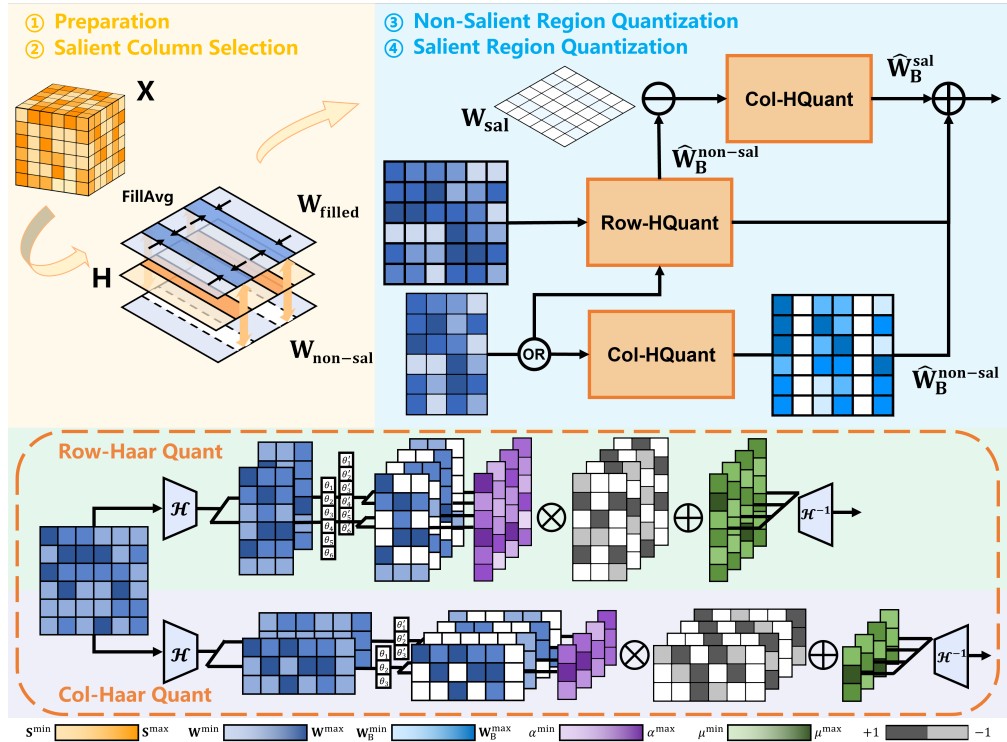

Figure 2: Overview of our HBLLM. The HBLLM quantization process consists of four steps: preparation, salient column selection, haar transform and quantization for the non-salient part, and quantization for the salient part. Since the salient columns are excluded from the Haar transform of the non-salient part, their positions must be filled before performing row-wise Haar transforms. This is handled by a process we refer to as FillAvg, where each missing column is filled with the average of its adjacent non-salient columns. For the non-salient part, HBLLM supports flexible choice between row-wise (HBLLM-row) and column-wise (HBLLM-col) transforms. The salient part undergoes column-wise Haar transformation followed by HaarQuant for quantization.

non-salient parts are quantized in the Haar domain, and their respective quantized Haar coefficients are denoted by $\widehat{\mathbf{W}}_{\mathrm{B}}^{\mathrm{sal}}$ and $\widehat{\mathbf{W}}_{\mathrm{B}}^{\mathrm{non\text{-}sal}}$. These are then reconstructed using inverse Haar transforms $\mathcal{H}_1^{-1}$ and $\mathcal{H}_2^{-1}$.

The reconstruction objective of HBLLM is twofold. For the quantization of a matrix layer $\mathbf{W}$, the objective expressed in the Frobenius norm is formulated as:

$$\min_{\widehat{\mathbf{W}}} \left\| \mathbf{W}\mathbf{X} - \widehat{\mathbf{W}}\mathbf{X} \right\|_F^2, \tag{1}$$

where $\mathbf{X}$ is the input of the matrix layer. For quantization of a matrix block $\mathbf{W}_{\mathrm{FP}}$ of $\mathbf{W}$, the object is:

$$\min_{\mathbf{M}_{\mathrm{sal}}, \widehat{\mathbf{W}}_{\mathrm{B}}^{\mathrm{sal}}, \widehat{\mathbf{W}}_{\mathrm{B}}^{\mathrm{non\text{-}sal}}} \left\| \mathbf{W}_{\mathrm{FP}} - \mathbf{M}_{\mathrm{sal}}\mathcal{H}_1^{-1}\left(\widehat{\mathbf{W}}_{\mathrm{B}}^{\mathrm{sal}}\right) - (\mathbf{I} - \mathbf{M}_{\mathrm{sal}})\mathcal{H}_2^{-1}\left(\widehat{\mathbf{W}}_{\mathrm{B}}^{\mathrm{non\text{-}sal}}\right) \right\|_F^2. \tag{2}$$

When $\mathcal{H}_1 = \mathcal{H}_2$ are fixed Haar transforms, this formulation simplifies to a quantization problem entirely in the Haar domain. In this case, the objective is the same to that of BiLLM. Layer-level quantization is commonly tackled with the GPTQ algorithm [12].

We emphasize that our approach does not aim to solve this objective function via explicit optimization. Instead, this formulation serves as a conceptual framework that guides our method design. The actual quantization process is based on a set of heuristics and structure-aware strategies that approximate this objective in a computationally efficient and scalable manner.

**Quantization Pipeline Overview.** HBLLM integrates the Haar transform into a BiLLM-style quantization pipeline (see Algorithm 1 and Figure 2), consisting of the following key steps:

1. **Preparation Phase:** Compute the column-wise importance scores using a Hessian-based saliency metric.

2. **Salient Column Selection and Quantization(SALIENT):**
   - Sort columns by their $\ell_2$ norm significance.
   - Select top-$K$ salient columns and determine $\mathbf{M}_{\text{sal}}$.
   - $\widehat{\mathbf{W}}_{\text{B}}^{\text{sal}} = \text{HaarQuant}\left(\mathbf{M}_{\text{sal}}\mathbf{W}\right)$.
   - Choose the subset with the lowest quantization error.

3. **Non-Salient Region Quantization:**
   - Fill the missing values in salient columns using adjacent averages (FillAvg).
   - $\widehat{\mathbf{W}}_{\text{B}}^{\text{non-sal}} = \text{HaarQuant}\left(\mathbf{M}_{\text{sal}}\mathbf{W}_{\text{filled}} + \left(\mathbf{I} - \mathbf{M}_{\text{sal}}\right)\mathbf{W}\right)$, where $\mathbf{W}_{\text{filled}}$ is from FillAvg.

4. **Adjustment and Refinement:**
   - $\widetilde{\mathbf{W}} = \mathbf{M}_{\text{sal}}\left(\mathbf{W} - \mathcal{H}^{-1}\left(\widehat{\mathbf{W}}_{B}^{\text{non-sal}}\right)\right)$.
   - $\widehat{\mathbf{W}}_{\text{B}}^{\text{sal}} = \text{HaarQuant}\left(\widetilde{\mathbf{W}}\right)$.

---

**Algorithm 1** Framework of HBLLM: Details of each function are shown in Algorithm E.1

---

**func** HBLLM($\mathbf{W}, \mathbf{X}, \beta, \lambda$)
**Input:** $\mathbf{W} \in \mathbb{R}^{n \times m}$ - weight matrix
        $\mathbf{X} \in \mathbb{R}^{r \times d}$ - calibration data
        $\beta$ - block size
        $\lambda$ - hessian regularizer
**Output:** $\mathbf{B}$ - haared binarized weights
1: $\mathbf{H} \leftarrow 2\mathbf{X}\mathbf{X}^{\top}$    // $\ell^2$ error hessian matrix
2: $\mathbf{H}^c \leftarrow \text{Cholesky}((\mathbf{H} + \lambda\mathbf{I})^{-1})$
3: $\mathbf{B} \leftarrow \mathbf{0}_{n \times m}$
4: **for** $b = 0, \beta, 2\beta, \ldots, N$ **do**
5:     $\mathbf{W}^b \leftarrow \mathbf{W}_{:,b:b+\beta}$
6:     $\text{rows}\{\cdot\} \leftarrow \text{SALIENT}(\mathbf{W}_{:,b:b+\beta}, \mathbf{H}^c)$
7:     **if** Row-HBLLM **then**
8:         $\mathbf{B}_{:,b:b+\beta} \leftarrow \text{Row-HaarQuant}(\mathbf{W}^b, \text{rows}\{\cdot\})$
9:     **else if** Col-HBLLM **then**
10:        $\mathbf{B}_{:,b:b+\beta} \leftarrow \text{Col-HaarQuant}(\mathbf{W}^b, \text{rows}\{\cdot\})$
11:     $\mathbf{E} \leftarrow (\mathbf{W}_{:,b:b+\beta} - \mathbf{B}_{:,b:b+\beta})/\mathbf{H}^c_{b:b+\beta,b:b+\beta}$
12:     $\mathbf{W}_{:,b+\beta:} \leftarrow \mathbf{W}_{:,b+\beta:} - \mathbf{E} \cdot \mathbf{H}^c_{b:b+\beta,b+\beta:}$
13: **return** $\mathbf{B}$

**func** Row-HaarQuant($\mathbf{W}, \text{rows}\{\cdot\}$)
1: $\mathbf{W}_{\text{filled}} \leftarrow \text{FillAvg}(\mathbf{W}_{:,j\notin\text{rows}}, \text{rows}\{\cdot\})$
2: $\mathbf{B}_{\text{filled}} \leftarrow \text{HaarQuant}(\mathbf{W}_{\text{filled}}, \text{ROW})$
3: $\widehat{\mathbf{W}} \leftarrow \mathbf{W} - \mathbf{B}_{\text{filled}}$
4: $\mathbf{B}_{\text{salient}} \leftarrow \text{HaarQuant}(\widehat{\mathbf{W}}_{:,j\in\text{rows}}, \text{COL})$
5: $\mathbf{B} \leftarrow \mathbf{B}_{\text{salient}} + \mathbf{B}_{\text{filled}}$
6: **return** $\mathbf{B}$

**func** Col-HaarQuant($\mathbf{W}, \text{rows}\{\cdot\}$)
1: $\mathbf{B}_{\text{unsalient}} \leftarrow \text{HaarQuant}(\mathbf{W}_{:,j\notin\text{rows}}, \text{COL})$
2: $\widehat{\mathbf{W}} \leftarrow \mathbf{W} - \mathbf{B}_{\text{filled}}$
3: $\mathbf{B}_{\text{salient}} \leftarrow \text{HaarQuant}(\widehat{\mathbf{W}}_{:,j\in\text{rows}}, \text{COL})$
4: $\mathbf{B} \leftarrow \mathbf{B}_{\text{salient}} + \mathbf{B}_{\text{unsalient}}$
5: **return** $\mathbf{B}$

---

### 3.3 HaarQuant: One-Bit Quantization in the Wavelet Domain

To boost expressiveness, we apply Haar wavelet transform to the weight matrix of linear layers, generating a frequency-domain coefficient matrix, followed by group-wise 1-bit quantization.

To address limited numerical expressiveness, HBLLM introduces the HaarQuant algorithm. Haar-Quant consists of three stages.

**Haar Transform.** A row of weights $\mathbf{W}$ is decomposed into low- and high-frequency coefficients via 1D Haar transform $\mathcal{H}$:

$$\widehat{\mathbf{W}} = \mathcal{H}\left(\mathbf{W}\right) = \left[\mathcal{H}_{\text{low-pass}}\left(\mathbf{W}\right), \mathcal{H}_{\text{high-pass}}\left(\mathbf{W}\right)\right], \tag{3}$$

where $\widehat{\mathbf{W}}$ is the Haar coefficient of $\mathbf{W}$, $\mathcal{H}_{\text{low-pass}}\left(\mathbf{W}\right)$ and $\mathcal{H}_{\text{high-pass}}\left(\mathbf{W}\right)$ are low- and high-frequency coefficients, respecively.

**Frequency-Aware Multi-Parameter Intra-Row Grouping.** For each row, boundary candidates determined by the row are enumerated, and the best grouping with minimal quantization error is selected. Furthermore, we split the rows by frequency bands. This adaptive strategy captures intra-row structural differences better than global uniform boundaries used in BiLLM.

**Coefficient Quantization.** Each group $\widehat{\mathbf{W}}_{\mathrm{FP}}$ is quantized using sign-based binarization centered on its mean:

$$\widehat{\mathbf{W}}_{\mathrm{B}} = \alpha \cdot \mathrm{sign}(\widehat{\mathbf{W}}_{\mathrm{FP}} - \mu), \tag{4}$$

where $\alpha \in \mathbb{R}^d$ is the row-wise scaling factor and $\mu$ is the group-wise mean and $\widehat{\mathbf{W}}_{\mathrm{B}}$ is the result.

## 3.4 Structure-aware Grouping Strategies

To enhance the fidelity and adaptability of binary quantization under structural constraints, HBLLM introduces two structure-aware grouping strategies that operate along both column and row dimensions of the weight matrix.

**Saliency-Driven Column Selection via $\ell_2$ Norm.** This strategy is used during salient column identification and quantization to overcome the limitations of prior heuristics based on fixed thresholds or simple magnitude criteria.

- Columns are ranked by their $\ell_2$-norm scores, which correlate with their overall contribution to activation magnitude.
- The top-$K$ columns are selected as salient and quantized in the Haar-transformed domain using column-wise transforms.

This approach helps preserve activation-critical directions, especially those dominated by outlier weights.

**Frequency-Aware Multi-Parameter Intra-Row Grouping.** This strategy is used during Haar domain quantization, where conventional row grouping lacks sensitivity to structural variations in weight distributions.

- Each row is first decomposed into high- and low-frequency components based on Haar subbands.
- Within each frequency band, coefficients are adaptively split into dense and sparse groups using band-specific, data-driven thresholds.

This grouping effectively doubles the number of quantization subgroups per row, enabling finer granularity and better error control.

Together, these strategies facilitate fine-grained, structure-preserving quantization across both dimensions of the weight matrix. To further guide saliency-based partitioning, we adopt the parameter importance metric used in BiLLM, defined as: $s_i = w_i^2 / [\mathbf{H}^{-1}]_{ii}^2$, where $\mathbf{H}$ denotes the Hessian matrix of the layer, $w_i$ is the full-precision value of the $i$-th parameter, and $[\mathbf{H}^{-1}]_{ii}$ is the $i$-th diagonal entry of the inverse Hessian.

This metric reflects the relative sensitivity of the loss to changes in each parameter: higher values indicate greater influence on the model's output, and thus prioritize that weight for accurate reconstruction.

## 3.5 Intra-frequency-band Mean Sharing

To reduce storage overhead, HBLLM shares a single mean value among 2 groups in the same frequency band within each row: $\mu_{\mathrm{shared}} = \frac{1}{n_1 + n_2} \left( \sum_{i=1}^{n_1} x_i + \sum_{j=1}^{n_2} y_j \right)$. It not only reduces per-parameter storage by 0.25 bits, but also maintains accuracy even slightly improving downstream task performance. This optimization achieves a trade-off between compression and accuracy, improving deployment viability.

## 3.6 Efficient Haar Implementation via Local Convolutions

Instead of costly matrix multiplication, HBLLM implements Haar transform using fixed local convolutions. There are only two predefined 1D kernels, $[1/2, 1/2]$ and $[1/2, -1/2]$, whose kernel size is 2. Furthermore, it can be hardcoded into the model for zero runtime initialization and no training or storage is needed. In complexity comparison, HBLLM needs $\mathcal{O}(d)$ operatons via convolutional sliding window, while FrameQuant needs $\mathcal{O}(d^2)$ operations. As a result, HBLLM significantly lowers inference cost and is ideal for edge deployment.

Table 1: Comparison of perplexity and average accuracy across models and methods

**LLaMA1**

| Size | Method | W-bits | C4 | Wiki2 | PTB | AvgQA↑ |
|------|--------|--------|------|-------|------|--------|
| | FullPrecision | 16.00 | 6.71 | 5.68 | 35.80 | 65.62 |
| 7B | FrameQuant | 2.20 | 10.89 | 9.96 | 104.7 | 56.19 |
| | PB-LLM | 1.70 | 90.19 | 113.4 | 830.0 | 35.71 |
| | BiLLM | 1.09 | 43.74 | 44.85 | 369.3 | 40.01 |
| | ARB-LLM$_X$ | 1.09 | 22.80 | 24.70 | 240.5 | 45.65 |
| | ARB-LLM$_{RC}$ | 1.09 | 15.13 | 13.45 | 155.8 | 52.23 |
| | HBLLM-row | 1.09 | **9.49** | **8.82** | **88.86** | **57.48** |
| | HBLLM-col | 1.00 | 10.38 | 9.67 | 117.7 | 54.03 |
| | FullPrecision | 16.00 | 6.24 | 5.09 | 25.36 | 68.09 |
| 13B | FrameQuant | 2.20 | 8.79 | 7.84 | 50.69 | 60.69 |
| | PB-LLM | 1.70 | 38.41 | 46.02 | 190.2 | 40.39 |
| | BiLLM | 1.10 | 13.93 | 14.99 | 69.75 | 50.89 |
| | ARB-LLM$_X$ | 1.10 | N/A | N/A | N/A | N/A |
| | ARB-LLM$_{RC}$ | 1.10 | 10.68 | 10.19 | 43.85 | 59.58 |
| | HBLLM-row | 1.09 | **7.62** | **6.68** | **34.94** | **62.57** |
| | HBLLM-col | 1.00 | 7.77 | 6.98 | 37.62 | 61.25 |
| | FullPrecision | 16.00 | 5.62 | 4.10 | 21.35 | 71.06 |
| 30B | FrameQuant | 2.20 | 7.35 | 6.32 | 28.69 | 65.13 |
| | PB-LLM | 1.70 | 21.73 | 25.87 | 127.1 | 47.22 |
| | BiLLM | 1.11 | 10.27 | 10.55 | 41.76 | 58.07 |
| | ARB-LLM$_X$ | 1.11 | N/A | N/A | N/A | N/A |
| | ARB-LLM$_{RC}$ | 1.11 | 8.49 | 7.79 | 30.98 | 64.49 |
| | HBLLM-row | 1.10 | **6.88** | **5.82** | **25.95** | **66.76** |
| | HBLLM-col | 1.00 | 7.03 | 6.03 | 26.65 | 64.86 |
| | FullPrecision | 16.00 | 5.31 | 3.53 | 21.11 | 72.27 |
| 65B | FrameQuant | 2.20 | 6.69 | 5.55 | 27.48 | 68.58 |
| | PB-LLM | 1.70 | 12.66 | 12.76 | 99.67 | 62.48 |
| | BiLLM | 1.10 | 9.26 | 8.58 | 41.93 | 62.05 |
| | ARB-LLM$_X$ | 1.10 | N/A | N/A | N/A | N/A |
| | ARB-LLM$_{RC}$ | 1.10 | 7.48 | 6.47 | 29.14 | 68.53 |
| | HBLLM-row | 1.09 | **6.28** | **5.07** | **24.11** | **69.18** |
| | HBLLM-col | 1.00 | 6.44 | 5.26 | 30.38 | 67.83 |

**LLaMA2**

| Size | Method | W-bits | C4 | Wiki2 | PTB | AvgQA↑ |
|------|--------|--------|------|-------|------|--------|
| | FullPrecision | 16.00 | 8.66 | 6.94 | 37.86 | 65.54 |
| 7B | FrameQuant | 2.20 | 14.66 | 13.34 | 177.1 | 52.75 |
| | PB-LLM | 1.70 | 63.95 | 55.40 | 486.2 | 36.54 |
| | BiLLM | 1.08 | 33.97 | 31.38 | 373.0 | 42.11 |
| | ARB-LLM$_X$ | 1.08 | 26.55 | 21.74 | 314.2 | 45.41 |
| | ARB-LLM$_{RC}$ | 1.08 | 17.87 | 15.85 | 462.2 | 46.71 |
| | HBLLM-row | 1.07 | **11.75** | **10.52** | **89.23** | **57.74** |
| | HBLLM-col | 1.00 | 12.51 | 11.33 | 150.6 | 54.09 |
| | FullPrecision | 16.00 | 6.18 | 4.88 | 43.02 | 69.18 |
| 13B | FrameQuant | 2.20 | 9.40 | 7.80 | 109.3 | 61.35 |
| | PB-LLM | 1.70 | 313.4 | 289.4 | 934.4 | 32.91 |
| | BiLLM | 1.08 | 22.17 | 19.57 | 303.4 | 46.76 |
| | ARB-LLM$_X$ | 1.08 | N/A | N/A | N/A | N/A |
| | ARB-LLM$_{RC}$ | 1.08 | 11.90 | 10.98 | 151.8 | 57.35 |
| | HBLLM-row | 1.07 | **7.82** | **6.71** | **61.75** | **63.61** |
| | HBLLM-col | 1.00 | 8.28 | 7.00 | 69.74 | 62.04 |
| | FullPrecision | 16.00 | 5.24 | 3.32 | 21.49 | 72.96 |
| 70B | FrameQuant | 2.20 | N/A | N/A | N/A | N/A |
| | PB-LLM | 1.70 | N/A | N/A | N/A | 54.26 |
| | BiLLM | 1.09 | 15.57 | 15.86 | 71.03 | 55.81 |
| | ARB-LLM$_X$ | 1.09 | N/A | N/A | N/A | N/A |
| | ARB-LLM$_{RC}$ | 1.09 | 7.26 | 6.00 | 28.43 | 68.77 |
| | HBLLM-row | 1.08 | **6.18** | **4.82** | **24.69** | **70.01** |
| | HBLLM-col | 1.00 | 6.63 | 5.04 | 26.31 | 68.61 |

**LLaMA3**

| Size | Method | W-bits | C4 | Wiki2 | PTB | AvgQA↑ |
|------|--------|--------|------|-------|------|--------|
| | FullPrecision | 16.00 | 11.90 | 8.29 | 13.07 | 68.94 |
| 8B | FrameQuant | 2.20 | 28.44 | 23.36 | 40.33 | 52.27 |
| | PB-LLM | 1.70 | 111.7 | 141.5 | 171.1 | 36.83 |
| | BiLLM | 1.06 | 53.67 | 56.24 | 81.27 | 41.84 |
| | ARB-LLM$_X$ | 1.06 | 48.45 | 37.90 | 52.59 | 43.40 |
| | ARB-LLM$_{RC}$ | 1.06 | 34.44 | 30.24 | 45.23 | 49.08 |
| | HBLLM-row | 1.06 | **20.09** | **16.18** | **22.83** | **54.80** |
| | HBLLM-col | 1.00 | 22.18 | 17.80 | 26.38 | 51.43 |
| | FullPrecision | 16.00 | 6.61 | 2.85 | 7.74 | 74.62 |
| 70B | FrameQuant | 2.20 | N/A | N/A | N/A | N/A |
| | PB-LLM | 1.70 | 33.56 | 28.93 | 44.38 | 47.45 |
| | BiLLM | 1.09 | 385.8 | 137.6 | 129.5 | 34.18 |
| | ARB-LLM$_X$ | 1.09 | N/A | N/A | N/A | N/A |
| | ARB-LLM$_{RC}$ | 1.09 | 12.80 | 10.24 | 12.76 | **63.90** |
| | HBLLM-row | 1.08 | **10.87** | **8.08** | **11.44** | 56.45 |
| | HBLLM-col | 1.00 | 13.69 | 9.09 | 14.26 | 55.89 |

**OPT**

| Size | Method | W-bits | C4 | Wiki2 | PTB | AvgQA↑ |
|------|--------|--------|------|-------|------|--------|
| | FullPrecision | 16.00 | 13.45 | 14.62 | 16.41 | 52.54 |
| 1.3B | FrameQuant | 2.20 | 24.29 | 27.15 | 30.45 | 44.48 |
| | PB-LLM | 1.70 | 186.9 | 309.0 | 286.3 | 33.44 |
| | BiLLM | 1.09 | 56.24 | 68.43 | 119.2 | 38.39 |
| | ARB-LLM$_X$ | 1.09 | 43.23 | 53.55 | 67.96 | 41.42 |
| | ARB-LLM$_{RC}$ | 1.09 | 24.23 | 28.77 | 33.32 | 45.28 |
| | HBLLM-row | 1.07 | **19.30** | **21.68** | **25.34** | **46.35** |
| | HBLLM-col | 1.00 | 21.92 | 24.08 | 27.28 | 44.70 |
| | FullPrecision | 16.00 | 12.06 | 12.47 | 14.61 | 54.95 |
| 2.7B | FrameQuant | 2.20 | 17.86 | 18.24 | 22.60 | **49.58** |
| | PB-LLM | 1.70 | 165.1 | 216.8 | 160.4 | 37.62 |
| | BiLLM | 1.10 | 42.92 | 55.75 | 103.2 | 40.02 |
| | ARB-LLM$_X$ | 1.10 | 30.02 | 34.15 | 41.35 | 44.60 |
| | ARB-LLM$_{RC}$ | 1.10 | 18.02 | 19.53 | 24.46 | 49.53 |
| | HBLLM-row | 1.09 | **15.70** | **16.85** | **19.54** | 48.80 |
| | HBLLM-col | 1.00 | 17.28 | 18.80 | 22.63 | 48.56 |
| | FullPrecision | 16.00 | 10.68 | 10.86 | 12.73 | 58.95 |
| 6.7B | FrameQuant | 2.20 | 14.53 | 14.59 | 18.71 | 53.77 |
| | PB-LLM | 1.70 | 122.9 | 206.7 | 222.3 | 34.87 |
| | BiLLM | 1.06 | 39.96 | 54.91 | 90.10 | 37.40 |
| | ARB-LLM$_X$ | 1.11 | 19.39 | 19.50 | 24.78 | 49.79 |
| | ARB-LLM$_{RC}$ | 1.11 | 14.29 | 15.16 | 17.92 | 53.76 |
| | HBLLM-row | 1.10 | **12.56** | **13.04** | **15.26** | **56.17** |
| | HBLLM-col | 1.00 | 13.29 | 13.67 | 15.70 | 54.44 |
| | FullPrecision | 16.00 | 10.16 | 10.13 | 11.89 | 58.41 |
| 13B | FrameQuant | 2.20 | 12.26 | 12.51 | 14.59 | 55.42 |
| | PB-LLM | 1.70 | 42.89 | 81.02 | 94.98 | 39.50 |
| | BiLLM | 1.13 | 17.01 | 18.34 | 21.56 | 49.82 |
| | ARB-LLM$_X$ | 1.13 | N/A | N/A | N/A | N/A |
| | ARB-LLM$_{RC}$ | 1.13 | 12.60 | 13.14 | 15.14 | 55.35 |
| | HBLLM-row | 1.12 | **11.47** | **11.72** | **13.78** | **55.91** |
| | HBLLM-col | 1.00 | 11.71 | 12.34 | 14.13 | 55.66 |
| | FullPrecision | 16.00 | 9.60 | 9.56 | 11.50 | 62.09 |
| 30B | FrameQuant | 2.20 | 10.92 | 11.15 | 13.25 | 59.62 |
| | PB-LLM | 1.70 | 21.60 | 28.62 | 45.63 | 46.14 |
| | BiLLM | 1.06 | 13.43 | 13.44 | 16.66 | 54.22 |
| | ARB-LLM$_X$ | 1.06 | N/A | N/A | N/A | N/A |
| | ARB-LLM$_{RC}$ | 1.06 | 11.18 | 10.94 | 13.27 | 58.59 |
| | HBLLM-row | 1.06 | **10.41** | **10.13** | **12.58** | **60.04** |
| | HBLLM-col | 1.00 | 10.53 | 10.29 | 12.75 | 58.91 |

*Note:* All methods are calibrated on C4 with 128 samples and a sequence length of 2048. A block size of 128 is used for channel-wise quantization, as commonly done in prior work. N/A: ARB-LLM$_X$ method cannot run on a single 3090 GPU - 24GB. W-bits is the average weight overhead per weight. For more details, please refer to the appendix D.

# 4 Experiments

## 4.1 Experimental Settings

**Models and Evaluation Datasets.** In our study, we evaluate HBLLM on various models, including those from the OPT, LLaMA-1, LLaMA-2, and LLaMA-3, as well as the recently introduced f-R1-Distill-Llama-8B. Specifically, we utilize the OPT models with 1.3B and 2.7B parameters, the LLaMA-1 and LLaMA-2 models with 7B and 13B parameters for our evaluations, and the LLaMA-3 model with 8B parameters. We measure language modeling capabilities of these models by evaluating their perplexity on the C4[26], WikiText2[22] and PTB[21] datasets. Additionally, we assess zero-shot accuracy on various Common Sense Reasoning Tasks such as PIQA[4], BoolQ[7], OpenBookQA[23], WinoGrande[28], ARC-e, ARC-c[8], HellaSwag[36], which are commonly used for evaluating the performance of LLM quantization methods. To further enhance evaluation coverage, we also include COPA[27] for causal reasoning and LAMBADA[25] for long-context language modeling. All evaluations are conducted using the open-source LLM evaluation framework, LM-Evaluation-Harness[24].

**Details of Experiments.** All experiments are conducted with PyTorch on NVIDIA GeForce RTX 3090 GPUs with 24GB of memory. For the calibration data, we follow the settings adopted in GPTQ and BiLLM, selecting 128 samples from the C4 dataset, with a sequence length of 2048. During quantization, we set the block size to 128 in BiLLM, PB-LLM, ARB-LLM, and HBLLM. Activations are kept in full precision (FP16).

**Baselines.** We compare HBLLM against several state-of-the-art LLM binarization methods, including BiLLM, ARB-LLM and PB-LLM, ensuring that all implementations adhere to the details provided in their respective papers. BiLLM, ARB-LLM and PB-LLM all utilize the PTQ approach for model calibration through OBQ based method of GPTQ. For ARB-LLM, we evaluate two of its best-performing variants, ARB-LLM$_X$ and ARB-LLM$_{RC}$. Both ARB-LLM$_x$ and ARB-LLM$_{RC}$ employ the salient column bitmap and group bitmap (CGB) for better performance. For PB-LLM, which allows variable ratios of salient weights to enhance accuracy, we have set the ratio of salient weights to 10% to ensure the average bit width of weight parameters remains below 2 bits. Given the significant accuracy improvements demonstrated by HBLLM over traditional binarization techniques, we also include a comparison with a leading method using orthogonal transforms: FrameQuant. For FrameQuant, quantization is performed not in the original weight space but in the structured orthogonal basis constructed through Fusion Frames. We evaluate two configurations: FrameQuant $(r = 1.0)$ and FrameQuant $(r = 1.1)$, where the redundancy factor r controls the amount of redundancy introduced during the transformation.

## 4.2 Perplexity and Accuracy Results of 1–2 Bit Quantized Models

The perplexity and zero-shot accuracy results of previous 1-2 bit quantization methods and the proposed HBLLM are presented in Table1. HBLLM consistently outperforms existing 1-2 bit quantization techniques across all evaluation metrics.

Specifically, HBLLM reduces the language modeling perplexity by 33%-66% compared to previous methods, while achieving substantial improvements in QA task accuracy, with relative gains ranging from $-0.73\%$ to $+11.3\%$. In our experiments, HBLLM slightly outperforms FrameQuant, a 2.2-bit quantization method, and exhibits a particularly significant advantage on the LLaMA-3-8B model. Moreover, when compared with BiLLM and ARB-LLM$_X$, HBLLM-col, demonstrates a clear advantage in both perplexity and accuracy, despite operating at comparable or lower bit-widths. These results indicate that HBLLM effectively narrows the performance gap between quantized models and their Float16 counterparts, achieving $1.22\times$ to $2.48\times$ of the original perplexity and retaining 73.8%-88.8% of the original QA accuracy.

## 4.3 Ablation Study

**Salient Column Selection Criterion.** To evaluate the impact of selection criteria in salient column screening on quantization effectiveness, we compare two strategies: the column $\ell_1$ norm and the column $\ell_2$ norm as significance indicators. Experimental results in Table 2a reveal that the column $\ell_2$ norm consistently achieves lower quantization error and superior performance in downstream tasks,

Table 2: Ablation study on LLaMA2-7B. Results are measured by perplexity, with final results highlighted in **bold**.

(a) Study of salient column selection criterion

| Method | Selection criterion | Wiki2↓ | PTB↓ |
|---|---|---|---|
| HBLLM-row | $\ell_1$ | 10.78 | 143.7 |
| | $\ell_2$ | **10.52** | **89.23** |
| HBLLM-col | $\ell_1$ | 11.45 | 308.2 |
| | $\ell_2$ | **11.33** | **150.6** |

(b) Study of grouping granularity

| Method | Group Partition | Wiki2↓ | PTB↓ |
|---|---|---|---|
| HBLLM-row | global | 16.32 | 1990 |
| | row-wise | **11.08** | **95.58** |
| HBLLM-col | global | 13.99 | 1546 |
| | row-wise | **12.02** | **146.1** |

(c) Effectiveness of shared mean

| Method | Shared mean | Wiki2↓ | PTB↓ |
|---|---|---|---|
| HBLLM-row | ✗ | 11.08 | 95.58 |
| | ✓ | **10.52** | **89.23** |
| HBLLM-col | ✗ | 12.02 | **146.1** |
| | ✓ | **11.33** | 150.6 |

(d) Study of partitioning candidates number

| Method | Candidate number | Wiki2↓ | PTB↓ |
|---|---|---|---|
| HBLLM-row | 10 | 11.16 | 108.8 |
| | 20 | 11.32 | 165.8 |
| | 40 | **11.08** | **95.58** |
| | 80 | 11.13 | 113.8 |

indicating its greater effectiveness in capturing energy distribution across columns and enhancing quantization quality.

**Granularity of Group Quantization.** To explore the influence of grouping granularity on model performance, we compare global grouping with row-wise grouping strategies, evaluating both quantization error and perplexity, as shown in Table 2b. The results reveal that row-wise grouping significantly reduces quantization error and achieves lower perplexity compared to global grouping. This suggests that finer-grained row-wise partitioning better preserves local data fidelity, leading to improved quantized inference performance.

**Shared Mean Strategy.** Under the standard dual-partition quantization setting, we further explore a compression strategy that shares the quantization center across two partitions within each row. By unifying the mean for both partitions, the storage overhead of quantization coefficients can be significantly reduced. Experimental results in Table 2c demonstrate that the shared mean strategy even slightly reduces quantization error without degrading perplexity, verifying its effectiveness.

**Choice of Partitioning Number.** We investigate the impact of varying the number of partition candidates on final quantization performance under the row-wise grouping setting. Specifically, for each row, we generate partition candidates based on absolute value percentiles ranging from 10% to 90%, and evaluate the corresponding quantization error and perplexity, as shown in Table 2d. Experimental results indicate that moderately increasing the number of partition candidates can effectively reduce quantization error and further lower perplexity, while excessive partitioning yields diminishing returns and increases computational cost. Consequently, we adopt 40 partition candidates as the default setting to balance performance and efficiency.

## 4.4 Time and Memory Analysis

**Time Comparison.** As a binary PTQ framework, HBLLM eliminates the need for finetuning. The introduction of Haar wavelet transforms requires additional computation during quantization, yet this overhead remains fully acceptable. As shown in Table 3, HBLLM increases the quantization time by approximately 20%-30% compared to BiLLM across different model sizes. It is worth noting that ARB-LLM$_X$ and FrameQuant fail to complete quantization for LLaMA-1-13B and LLaMA-1-30B under the single-GPU-24 GB setting, while HBLLM successfully completes the process, demonstrating better scalability.

Table 3: Time comparison between LLM binarization methods and our HBLLM on LLaMA-1 with different model sizes.

| Method | 7B | 13B | 30B |
|---|---|---|---|
| BiLLM | 36min | 71min | 142min |
| ARB-LLM$_X$ | 88min | ✗ | ✗ |
| ARB-LLM$_{RC}$ | 76min | 119min | 239min |
| PB-LLM | 18min | 29min | 57min |
| FrameQuant | 14min | 22min | ✗ |
| HBLLM | 44min | 98min | 173min |

**Memory Comparison.** As shown in Table 4, HBLLM-col achieves better performance while occupying a storage size comparable to ARB-LLM. By employing a grouped shared-mean strategy, HBLLM improves compression efficiency without sacrificing performance. Specifically, HBLLM-col applies Haar transforms along the column dimension, such that only one grouped quantization operation is required per row on the transformed coefficients. Compared to HBLLM-row, this leads to reduced data fidelity but provides clear advantages in storage cost. Notably, the reported memory usage is measured at runtime in our setup and may be influenced by model variants and implementation choices, leaving room for further engineering optimizations. The detailed storage calculation formulas can be found in the appendix D.

Table 4: Memory comparison LLM binarization methods and our HBLLM on LLaMA-1 with different model sizes.

| Method | 7B | 13B |
|---|---|---|
| FP16 | 13.48GB | 26.03GB |
| BiLLM | 2.93GB | 5.36GB |
| ARB-LLM$_x$ | 3.23GB | 5.95GB |
| ARB-LLM$_{RC}$ | 2.83GB | 5.17GB |
| PB-LLM | 2.91GB | 5.33GB |
| FrameQuant | 11.36GB | 16.08GB |
| HBLLM-row | 3.29GB | 6.07GB |
| HBLLM-col | 2.86GB | 5.22GB |

### 4.5 Inference Latency Estimation

To evaluate the inference latency of HBLLM, we conduct an experiment that combines direct measurement with estimation. Due to there is no existing inference framework that fully supports the dequantization algorithm used in HBLLM, we test GEMV on layers from the OPT-175B model instead. The tests are run on an NVIDIA P100 GPU following the GPTQ benchmark setup [1]. Our estimation results show that the inference latency of HBLLM is approximately $31.8\%$ of the FP16 baseline inference time. For more details, please refer to the appendix G.

## 5 Conclusion

We introduce a 1-bit weight only quantization HBLLM, which applies Haar transform to BILLM pipeline. Besides quantifying the coefficients on frequence domain, HBLLM integrates two innovative structure-aware grouping strategies to enhance fidelity. Furthermore, HBLLM optimize storage efficiency. As a results, HBLLM outperforms SOTA QAT quantization methods of LLM at 1-bit across different LLM families and tests. The current HBLLM supports only quantized dense models. Next, we will focus on the MoE PTQ algorithm.

## Acknowledgements

We thank all constructive comments from anonymous reviewers. This work is partially supported by the National Key Research and Development Program of China under Grant No.2023YFB3001704.

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

# A   Introducing Haar Wavelet Transform into LLM Quantization

Haar wavelet transform converts an original vector into a new matrix containing low- and high-frequency information by computing averages and differences. This transformation can be extended to larger vectors and multidimensional data and is commonly used in image processing and signal analysis.

Applying the Haar wavelet transform to LLM quantization offers three advantages:

- The inverse quantization set becomes richer in representation, which can be demonstrated via the CIQ metric.

- The data distribution becomes more concentrated: with approximately 65% probability, the variance of the high- and low-frequency coefficient sets in each row is smaller than before the transformation.

- The additional computational cost for inference is $\mathcal{O}(d)$, where $d$ is the input length, as the Haar transform can be implemented using local convolutional layers, resulting in lower cost than methods such as FrameQuant.

## A.1   Definition of Haar Transform

Let the one-dimensional input signal be $x$. The output after wavelet transformation is:

$$\widehat{x} := [a_1, b_1, \ldots, a_i, b_i], \tag{5}$$

where $a_i$ denotes low-frequency coefficients and $b_i$ denotes high-frequency coefficients.

This process can also be expressed in matrix form:

$$\widehat{x}^T = \mathcal{H}(x) := \mathbf{U}_{\text{diag}} x^T, \tag{6}$$

where

$$\mathbf{U}_{\text{diag}} := \begin{pmatrix} \mathbf{U} & & & \\ & \mathbf{U} & & \\ & & \ddots & \\ & & & \mathbf{U} \end{pmatrix}. \tag{7}$$

Because the Haar matrix is orthogonal, the inverse Haar transform is:

$$x^T = \mathbf{U}_{\text{diag}}^T \widehat{x}^T. \tag{8}$$

### A.1.1   Row-wise and Column-wise Haar Transforms on Matrices

Define row-wise and column-wise Haar transforms of a matrix $\mathbf{W}$ as:

$$\mathcal{H}_{\text{row}}(\mathbf{W}) := \mathbf{W}\mathbf{U}_{\text{diag}}^T, \quad \mathcal{H}_{\text{col}}(\mathbf{W}) := \mathbf{U}_{\text{diag}}\mathbf{W}. \tag{9}$$

Below is a simple example showing how to apply Haar transforms to a matrix.

Given a $4 \times 4$ matrix defined as:

$$\mathbf{A} = \begin{pmatrix} 16 & 18 & 22 & 20 \\ 12 & 14 & 10 & 8 \\ 24 & 26 & 30 & 28 \\ 20 & 22 & 18 & 16 \end{pmatrix}, \tag{10}$$

the result after applying row-wise Haar transform is denoted as

$$\widehat{\mathbf{A}}_{\text{row}} = \sqrt{2} \times \begin{pmatrix} 17 & -1 & 21 & 1 \\ 13 & -1 & 9 & 1 \\ 25 & -1 & 29 & 1 \\ 21 & -1 & 17 & 1 \end{pmatrix}, \tag{11}$$

and after applying column-wise Haar transform is denoted as

$$\widehat{\mathbf{A}}_{\text{col}} = \sqrt{2} \times \begin{pmatrix} 15 & 0 & 23 & 0 \\ -2 & 0 & -2 & 0 \\ 15 & 0 & 23 & 0 \\ -2 & 0 & -2 & 0 \end{pmatrix}. \tag{12}$$

where each element of matrices is multiplied by the coefficient $\sqrt{2}$.

# B Relationship Between Richness of Inverse Quantization Set and Model Fidelity

## B.1 Definition and Application of CIQ Metric

The cardinality of an inverse quantization set (CIQ) measures the number of distinct values that can be recovered from quantized weights in each row. For linear quantization without partitioning, CIQ equals the bit-width of quantized weights, since linear quantization evenly distributes several points across the original data range. When partitioning strategies are applied, it characterize expressiveness of the quantization algorithm.

In HBLLM, which follows the GPTQ quantization scheme, quantization is done per row of each matrix block. We define CIQ in terms of a single row of a matrix block in the following discussion without loss of generality. Furthermore,

$$CIQ = \min\{\text{row length, maximum recovery ability under given quantization parameters}\}.$$

holds.

To study the composition of IQ, we introduce a mapping from the set of quantized weights to the set of dequantized weights. Several mappings from quantized weights to inverse quantized values exist:

- Identity mapping
- Group merging mapping
- Residual merging mapping
- Inverse transformation mapping

For example, an inverse quantization set brought by residual merging mapping is defined as:

$$\mathbf{IQ}_{\text{residual}} = \text{inv}_{\text{residual}}\left(\mathbf{X}_1, \mathbf{X}_2\right) := \{z : z = (x + y), \forall x \in \mathbf{X}_1, \forall y \in \mathbf{X}_2\}. \tag{13}$$

Furthermore,

$$CIQ_{\text{residual}} \leq |\mathbf{X}_1| \cdot |\mathbf{X}_2|, \tag{14}$$

where $|\cdot|$ is the cardinality of a set.

An inverse quantization set brought by Haar inverse transformation is defined as:

$$\mathbf{IQ}_{\text{Haar}} = \text{inv}_{\text{Haar}}\left(\widehat{\mathbf{X}}_{\text{low}}, \widehat{\mathbf{X}}_{\text{high}}\right) := \left\{z : z = \frac{1}{\sqrt{2}}(x + y) \text{ or } z = \frac{1}{\sqrt{2}}(x - y), \forall x \in \widehat{\mathbf{X}}_{\text{low}}, \forall y \in \widehat{\mathbf{X}}_{\text{high}}\right\}. \tag{15}$$

And then,

$$CIQ_{\text{Haar}} \leq 2|\widehat{\mathbf{X}}_{\text{low}}| \cdot |\widehat{\mathbf{X}}_{\text{high}}|. \tag{16}$$

Let an inverse quantization set produced by BiLLM algorithm be denoted as $\mathbf{IQ}_{\text{BiLLM}}$. According to BiLLM algorithm, $\mathbf{IQ}_{\text{BiLLM}}$ can be expressed in the following form:

$$\begin{aligned}
\mathbf{IQ}_{\text{BiLLM}} &= \mathbf{IQ}_{\text{residual}}^{\text{sal}} \cup \mathbf{IQ}^{\text{non-sal}} \\
&= \text{inv}_{\text{residual}}\left(\mathbf{X}_1^{\text{sal}}, \mathbf{X}_2^{\text{sal}}\right) \cup \mathbf{X}_1^{\text{non-sal}} \cup \mathbf{X}_2^{\text{non-sal}}.
\end{aligned} \tag{17}$$

where $\mathbf{IQ}_{\text{residual}}^{\text{sal}}$ and $\mathbf{IQ}^{\text{non-sal}}$ represent inverse quantization sets of the salient and non-salient parts, respectively.

**Lemma 1.** *BiLLM has at most 8 different dequantized values per row.*

*Proof.* According to (17) and the definition of CIQ, we can infer that

$$CIQ_{\text{BiLLM}} \leq CIQ_{\text{residual}}^{\text{sal}} + |\mathbf{X}_1^{\text{non-sal}}| + |\mathbf{X}_2^{\text{non-sal}}|. \tag{18}$$

Since the salient part adopts a residual approximation strategy, it follows that

$$CIQ_{\text{residual}}^{\text{sal}} \leq |\mathbf{X}_1|^{\text{sal}} \cdot |\mathbf{X}_2|^{\text{sal}}. \tag{19}$$

Given that $|\mathbf{X}_1^{\text{sal}}| = |\mathbf{X}_2^{\text{sal}}| = |\mathbf{X}_1^{\text{non-sal}}| = |\mathbf{X}_2^{\text{non-sal}}| = 2$, substituting (19) into (18) yields:

$$CIQ_{\text{BiLLM}} \leq \left|\mathbf{X}_1^{\text{sal}}\right|\left|\mathbf{X}_2^{\text{sal}}\right| + \left|\mathbf{X}_1^{\text{non-sal}}\right| + \left|\mathbf{X}_2^{\text{non-sal}}\right| = 2 \times 2 + 2 + 2 = 8. \tag{20}$$

$\square$

Next, we analyze theoretical upper bounds of $CIQ$ for HBLLM algorithms. We first consider the case of HBLLM-col. Let an inverse quantization set produced by HBLLM-col be denoted as $\mathbf{IQ}_{\text{HBLLM-col}}$. According to HBLLM-col algorithm, $\mathbf{IQ}_{\text{HBLLM-col}}$ can be expressed in the following form:

$$\begin{aligned} \mathbf{IQ}_{\text{HBLLM-col}} &= \mathbf{IQ}_{\text{Haar}}^{\text{sal}} \cup \mathbf{IQ}_{\text{Haar}}^{\text{non-sal}} \\ &= \text{inv}_{\text{Haar}}\left(\widehat{\mathbf{X}}_{\text{low}}^{\text{sal}}, \widehat{\mathbf{X}}_{\text{high}}^{\text{sal}}\right) \cup \text{inv}_{\text{Haar}}\left(\widehat{\mathbf{X}}_{\text{low}}^{\text{non-sal}}, \widehat{\mathbf{X}}_{\text{high}}^{\text{non-sal}}\right), \end{aligned} \tag{21}$$

where $\mathbf{IQ}_{\text{Haar}}^{\text{sal}}$ and $\mathbf{IQ}_{\text{Haar}}^{\text{non-sal}}$ represent the dequantized sets of the salient and non-salient parts, respectively. Both parts employ a group quantization strategy under Haar transform, so the upper bound for each part is the product of the cardinality of the two Haar sub-band quantization sets. Additionally, within HBLLM-col algorithm, each sub-band has two groups, resulting in a total of four quantized values. Based on the above analysis, we arrive at the second conclusion.

**Lemma 2.** *HBLLM-col has at most* 64 *different dequantized values per row.*

*Proof.*

$$CIQ_{\text{HBLLM-col}} \le 2\left|\widehat{\mathbf{X}}_{\text{low}}^{\text{sal}}\right|\left|\widehat{\mathbf{X}}_{\text{high}}^{\text{sal}}\right| + 2\left|\widehat{\mathbf{X}}_{\text{low}}^{\text{non-sal}}\right|\left|\widehat{\mathbf{X}}_{\text{high}}^{\text{non-sal}}\right| = 64. \tag{22}$$

$\square$

The $CIQ$ upper bound of HBLLM-row algorithm is significantly larger than that of HBLLM-col algorithm. Let an inverse quantization set produced by HBLLM-row algorithm be denoted as $\mathbf{IQ}_{\text{HBLLM-row}}$. By HBLLM-row algorithm, $\mathbf{IQ}_{\text{HBLLM-row}}$ can be shown in the following form:

$$\mathbf{IQ}_{\text{HBLLM-row}} = \mathbf{IQ}_{\text{HBLLM-row}}^{\text{sal}} \cup \mathbf{IQ}_{\text{Haar}}^{\text{non-sal}}, \tag{23}$$

where $\mathbf{IQ}_{\text{HBLLM-row}}^{\text{sal}}$ is defined as

$$\mathbf{IQ}_{\text{HBLLM-row}}^{\text{sal}} = \text{inv}_{\text{residual}}\left(\mathbf{IQ}_{\text{Haar}}^{\text{sal}}, \mathbf{IQ}_{\text{Haar}}^{\text{non-sal}}\right). \tag{24}$$

Unlike HBLLM-col, the non-salient part of HBLLM-row encompasses the entire matrix area, resulting in overlap with the salient part. Therefore, HBLLM-row incorporates a residual approximation on the salient part, further increasing the $CIQ$ upper bound. Additionally, within HBLLM-row algorithm, each sub-band has two groups, resulting in a total of four quantized values. Based on the above analysis, we arrive at the third conclusion.

**Lemma 3.** *HBiLLM-row can have over* 1024 *different dequantized values per row.*

*Proof.*

$$\begin{aligned} CIQ_{\text{HBLLM-row}}^{\text{sal}} &\le \left|\mathbf{IQ}_{\text{Haar}}^{\text{sal}}\right|\left|\mathbf{IQ}_{\text{Haar}}^{\text{non-sal}}\right| \\ &= \left|\text{inv}_{\text{Haar}}\left(\widehat{\mathbf{X}}_{\text{low}}^{\text{sal}}, \widehat{\mathbf{X}}_{\text{high}}^{\text{sal}}\right)\right| \times \left|\text{inv}_{\text{Haar}}\left(\widehat{\mathbf{X}}_{\text{low}}^{\text{non-sal}}, \widehat{\mathbf{X}}_{\text{high}}^{\text{non-sal}}\right)\right| \\ &\le 2\left|\widehat{\mathbf{X}}_{\text{low}}^{\text{sal}}\right|\left|\widehat{\mathbf{X}}_{\text{high}}^{\text{sal}}\right| \times 2\left|\widehat{\mathbf{X}}_{\text{low}}^{\text{non-sal}}\right|\left|\widehat{\mathbf{X}}_{\text{high}}^{\text{non-sal}}\right| \\ &= 1024. \end{aligned} \tag{25}$$

$$CIQ_{\text{HBLLM-row}} \le 1024 + 32 = 1056. \tag{26}$$

$\square$

Lemma 3 is a result that holds under the assumption that there are sufficiently many columns in the salient part. In practical algorithms, due to the dual constraints of the quantized matrix size and the total bitrate, the upper bound of $CIQ_{\text{HBLLM-row}}$ is much less than 1024. The specific upper bound can be described in Lemma 4.

**Lemma 4.** *Let the size of a quantized matrix block be* $d \times d$*, where* $d \le 256$*, and let the proportion of the number of columns in the salient part to the total number of columns be* $p$ *(where* $0 < p < 1$*). Then, we have*

$$CIQ_{HBLLM\text{-}row} \le 32 + p \cdot d, \tag{27}$$

*and*

$$CIQ_{HBLLM\text{-}col} \le 32 + \min\{p \cdot d, 32\}. \tag{28}$$

*Proof.* It is easy to get by Lemma 2 and Lemma 3. $\square$

**Theorem 1.** *Under the same proportion* $p$ *of the salient part,* $CIQ_{HBLLM\text{-}row} \ge CIQ_{HBLLM\text{-}col}$ *holds.*

*Proof.* This follows from Lemma 4. $\square$

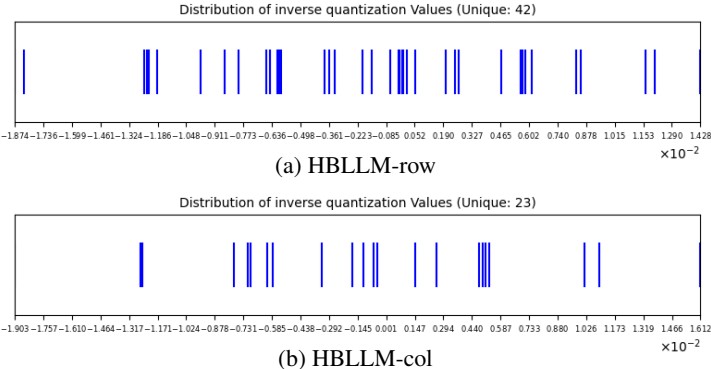

(a) HBLLM-row

(b) HBLLM-col

Figure B.1: CIQ and Distribution of inverse quantization Values

## B.2 Measured Values of the CIQ Metric for the HBLLM Algorithm

To validate the theoretical analysis of CIQ, we present two representative examples under the HBLLM-row and HBLLM-col schemes, respectively. These examples illustrate how the CIQ values measured in practice align with the theoretical bounds derived earlier.

**Example 1: HBLLM-row.** As shown in B.1a, consider a row of length 128 where 14 elements are marked as salient. After quantization and reconstruction, the total number of distinct values observed is 42. Among these, 30 values originate from the non-salient part (i.e., $|\mathbf{IQ}_{\text{Haar}}^{\text{non-sal}}| = 30$) and 7 values from the salient part (i.e., $|\mathbf{IQ}_{\text{Haar}}^{\text{sal}}| = 7$).

Although the salient part involves residual merging, the final reconstructed set still satisfies:

$$CIQ_{\text{HBLLM-row}} = 42 \leq 30 + 14 = 44, \tag{29}$$

which confirms that the practical CIQ value stays within the theoretical upper bound in Lemma 4.

**Example 2: HBLLM-col.** As shown in B.1b, In a second example under the HBLLM-col scheme, a row consists of 119 non-salient and 9 salient elements. The measured CIQ is 23, with 16 values from the non-salient part and 7 values from the salient part. This result again satisfies:

$$CIQ_{\text{HBLLM-col}} = 23 \leq 16 + 9 = 25. \tag{30}$$

These observations demonstrate that in practice, the effective size of the inverse quantization set is significantly below the worst-case bounds, especially when some quantized values are shared or overlap. They also confirm the effectiveness of the HBLLM decomposition strategies in maintaining a compact and expressive representation of quantized weights.

## B.3 Limitations of CIQ Metric Analysis and the Necessity of Introducing a Structure-Aware Grouping Strategy

Although HBLLM significantly outperforms BiLLM in the CIQ metric, it lacks a clear advantage in quantization performance without the structure-aware grouping strategy introduced in this paper.

To assess the performance differences before and after implementing this strategy, we conduct experiments evaluating perplexity and QA accuracy. Experiment data can be found in Table B.1:

- BiLLM+$\ell_2^\dagger$: Employing $\ell_2$-based saliency-driven column selection together with multi-parameter intra-row grouping.

- Haar+BiLLM: This refers to a method obtained by removing the $\ell_2$-norm-based saliency-driven column selection and multi-parameter intra-row grouping strategies from HBLLM. This method integrates the one-dimensional discrete Haar wavelet transform applied row-wise or column-wise, thereby deriving two approaches: Row-Haar+BiLLM and Col-Haar+BiLLM, respectively.

- DCT+BiLLM: This approach applies the BiLLM algorithm to coefficient matrices obtained from the one-dimensional Discrete Cosine Transform (DCT) applied row-wise or column-wise on weight matrices, resulting in Row-DCT+BiLLM and Col-DCT+BiLLM. Unlike the Haar+BiLLM method, DCT+BiLLM uses a global transformation strategy by first mapping the entire matrix to the Fourier domain before quantization. In contrast, Haar+BiLLM applies the Haar transform to matrix blocks, with quantization following the BillM process, making it a local orthogonal transformation.

- HBiLLM+: These method, refer to those derived from Haar+BiLLM, utilizing the strategies proposed in our paper. They include HBLLM-row+$\ell_2$, HBLLM-col+$\ell_2$, HBLLM-col+$\ell_2^\dagger$ and HBLLM-col+$\ell_2^\dagger$.

Table B.1: Perplexity ($\downarrow$, C4, Wiki2, PTB) and AvgQA accuracy ($\uparrow$, AvgQA over 9 zero-shot tasks) of BiLLM variants with Haar.

| Method | OPT-1.3B | | | | LLaMA2-7B | | | |
|---|---|---|---|---|---|---|---|---|
| | C4$\downarrow$ | Wiki2$\downarrow$ | PTB$\downarrow$ | AvgQA$\uparrow$ | C4$\downarrow$ | Wiki2$\downarrow$ | PTB$\downarrow$ | AvgQA$\uparrow$ |
| BiLLM | 56.24 | 68.43 | 119.2 | 38.39 | 33.97 | 31.38 | 373.0 | 42.11 |
| BiLLM+$\ell_2$ | 55.95 | 72.42 | 105.9 | 37.95 | 33.46 | 31.34 | 695.8 | 41.11 |
| BiLLM+$\ell_2^\dagger$ | 56.88 | 70.48 | 92.16 | 39.28 | 28.17 | 25.08 | 226.3 | 41.77 |
| Row-Haar+BiLLM | 47.45 | 52.81 | 62.81 | 39.57 | 25.77 | 25.12 | 138.0 | 44.67 |
| Col-Haar+BiLLM | 95.56 | 128.8 | 171.3 | 36.92 | 41.03 | 37.25 | 5193 | 39.60 |
| Row-DCT+BiLLM | 8010 | 11517 | 6729 | 31.36 | 45358 | 49395 | 26888 | 34.48 |
| Col-DCT+BiLLM | 107.1 | 150.5 | 250.1 | 34.19 | 26.54 | 24.64 | 1202 | 44.82 |
| HBLLM-row+$\ell_2$ | 26.47 | 33.68 | 41.17 | 41.17 | 16.26 | 19.86 | 87.90 | 47.90 |
| HBLLM-col+$\ell_2$ | 26.37 | 29.99 | 36.24 | 42.13 | 15.04 | 13.99 | 154.6 | 49.38 |
| HBLLM-row+$\ell_2^\dagger$ | **19.55** | 26.95 | **25.70** | 46.87 | **13.00** | 13.20 | **85.50** | 50.86 |
| HBLLM-col+$\ell_2^\dagger$ | 21.98 | **23.69** | 27.39 | **45.21** | 13.18 | **12.02** | 146.1 | **51.34** |

*Note:* $\ell_2$ denotes activation of $\ell_2$-norm-based saliency-driven column selection; $\dagger$ denotes activation of frequency-aware multi-parameter intra-row grouping.

The main experimental results are summarized as follows:

- Directly applying Haar transform into BiLLM pipeline does not significantly improve 1-bit quantization performance.
  - Row-Haar+BiLLM shows slight improvement.
  - Col-Haar+BiLLM decreases performance.
- Implementing the 'saliency-driven column selection via $\ell_2$ norm' strategy leads to:
  - Significant improvements for HBLLM-row+$\ell_2$ and HBLLM-col+$\ell_2$ compared to their predecessors.
  - Perplexity tests show:
    * 32-64% reductions on C4 and Wiki2 test sets.
    * HBLLM-col+$\ell_2$ shows notable improvements, but still lags behind BiLLM on the PTB test set.
  - QA testing accuracy improves by 3-10%.
- Further introducing 'frequency-aware multi-parameter intra-row grouping' results in:
  - HBLLM-row+$\ell_2$+Row-wise-grouping is the best.
  - 26-45% reductions in perplexity on C4 and Wiki2.
  - Significant improvements on the PTB test set, surpassing BiLLM.
  - Cumulative accuracy in QA testing increases by 2-8%.

This experimental result demonstrates that introducing a structure-aware grouping strategy is essential for effectively combining the Haar transform with the BiLLM algorithm.

## B.4 Effectiveness of Haar Transform and the Importance of Local Orthogonality

As shown in Table B.1, although HBLLM integrates multiple strategies, it is important to disentangle the specific contribution of the Haar-based frequency decomposition from other components such as saliency selection and structure-aware grouping. To this end, we conduct dedicated ablation studies to quantify the standalone effectiveness of the Haar transform and contrast it with global orthogonal alternatives such as Discrete Cosine Transform (DCT).

We summarize our key observations below:

- **Effectiveness of Haar Transform:** While incorporating either the $\ell_2$-based saliency selection or the structure-aware grouping alone yields only modest improvements to BiLLM, introducing the Haar transform leads to consistently more substantial gains in both perplexity and QA accuracy. Notably, even under partial activation (e.g., HBLLM-col+$\ell_2$), the models outperform their BiLLM counterparts.
  - Both HBLLM-row+$\ell_2^\dagger$ and HBLLM-col+$\ell_2^\dagger$ significantly outperform BiLLM and BiLLM+$\ell_2^\dagger$, underscoring the crucial role of Haar in preserving the frequency-domain structure of weights.
  - Row-Haar+BiLLM, even without $\ell_2$-norm-based saliency-driven column selection or grouping, shows consistent performance gains, confirming that Haar decomposition independently enhances quantization representation capacity.

- **Local vs. Global Orthogonal Transforms:** We further analyze the impact of replacing Haar transform with global DCT.
  - Applying the global row-wise DCT results in severe degradation across all benchmarks, with Row-DCT+BiLLM performing significantly worse than BiLLM.
  - Applying the global column-wise DCT offers moderate improvement on LLaMA2-7B; however, Col-DCT+BiLLM still lags behind BiLLM in all OPT-1.3B tests.
  - These results highlight that global transforms struggle to capture local variations in weight distributions, which are effectively preserved by block-wise Haar decomposition.

As shown in Table B.2, global transforms such as DCT also incur substantial computational overhead compared to local Haar transforms:

Table B.2: Time comparison between BiLLM, DCT+BiLLM, and HBLLM on LLaMA-1 with different model sizes. The DCT implementation used in this test is from pytorch.

| Method | 7B | 13B | 30B |
|---|---|---|---|
| BiLLM | 36 min | 71 min | 142 min |
| DCT+BiLLM | 211 min | 414 min | 1012 min |
| HBLLM | 44 min | 98 min | 173 min |

In conclusion: Haar transform independently and robustly contributes to quantization fidelity, even in the absence of auxiliary strategies such as $\ell_2$-norm-based saliency-driven column selection or grouping; local orthogonal transforms like Haar are consistently more effective than global ones like DCT in preserving localized frequency-domain structures—an essential property for stable and expressive 1-bit quantization.

## C  Analysis of the Correlation Between Data Concentration and Model Fidelity

In this section, we explores the positive correlation between improved data concentration and enhanced model fidelity after quantization, following the application of the Haar transform and structure-aware grouping strategy proposed by HBLLM. This correlation provides a theoretical foundation for the effectiveness of the HBLLM quantization method. We first observe that the concentration of coefficient distribution improves with a $65\%$ probability after applying the Haar row transform. Based on this observation, we mathematically model the probability of variance improvement in data concentration. We then apply this variance improvement probability to the HBLLM-row and HBLLM-col methods to validate the correlation between enhanced data concentration and improved model fidelity after quantization.

### C.1  Improvement of Data Concentration by Haar Transform

We discusses the improvement in data concentration resulting from the Haar transform, which is typically described by variance—lower variance indicates higher data concentration. We examine the impact of the Haar row transform on the distribution characteristics of the weight matrix. Specifically, we select a matrix block from the OPT-1.3B model and compare the variance of each original weight vector with the variances of the low-frequency and high-frequency subbands obtained after Haar decomposition. The variances of each row from different methods are arranged in ascending order, as shown in Figure C.1. Notably, approximately $65\%$ of the rows exhibit a variance in at least one subband that is lower than the original value, indicating that the Haar row transform generally enhances data concentration.

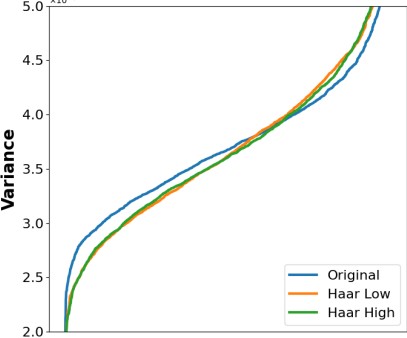

Figure C.1: Row-wise Variance Comparison Before and After Haar Transform

Therefore, we anticipate that the application of HBLLM's structure-aware grouping strategy may further improve data concentration.

## C.2    Mathematical Modeling of Variance Improvement Probability

To quantify the improvement in data concentration achieved by the Haar transform combined with the structure-aware grouping strategy, we introduce a random variable $p$ to describe the probability of improvement in data concentration for each row after applying the strategy, as well as the expected probability of improvement $E(p)$ for each matrix block. Let the matrix block size be $d \times d$, and the $j$-th row, after the Haar row transform, be divided into $M$ subgroups, each containing $n_i^j$ coefficients, with an intra-group variance of $V_i^j$. The variance of the entire row before transformation is denoted as $\tilde{V}_0$. The sample value $p_j$ for the $j$-th row is calculated as follows:

$$p_j := \frac{\sum_{i=1}^{M} n_i^j \times \text{sign}\Big(\max\{0, (\tilde{V}_0 - V_j)\}\Big)}{d}, \tag{31}$$

where, the sign function $\text{sign}(\cdot)$ takes the value of $1$ only when the subgroup variance is less than the original row variance; otherwise, it is $0$. The value $p_j$ represents the proportion of coefficients in the $j$-th row that belong to subgroups with improved concentration. Based on this, we define the expected value $E(p)$ for the entire matrix block to characterize the average level of overall data concentration improvement:

$$E(p) = \frac{\sum_{j=1}^{d} p_j}{d}. \tag{32}$$

## C.3    Analysis of the Relationship Between Data Concentration and Quantization Fidelity

To further investigate the impact of improved data concentration on model fidelity, we collected data on the probability of variance improvement, the relative $\ell_2$ error of matrix blocks before and after quantization, and the corresponding model fidelity, as shown in Figure C.2. The relative $\ell_2$ error serves as the optimization criterion for HBLLM quantization, while model fidelity is measured by perplexity—lower perplexity indicates higher fidelity after quantization.

We analyzed the distribution changes of relative $\ell_2$ errors for all matrix blocks across different models before and after quantization, and we plotted the perplexity performance under various grouping strategies. The experimental results demonstrate that HBLLM not only significantly enhances data concentration but also effectively mitigates the growth of quantization error, thereby better preserving the model's original performance.

Figure C.2a displays the proportion $E(p)$ of Haar coefficients across all matrix blocks that meet the variance improvement criterion after combining the row wavelet transform and grouping strategy. The points on the graph represent $E(p)$ for a matrix block in a quantized model, with each graph showing the values of $E(p)$ arranged in ascending order. As illustrated, after applying the strategy, over $65\%$ of Haar coefficients in all matrix blocks achieved an improvement in data concentration, with median improvements of $67\%$ and $68\%$, respectively.

The results from Figure C.2 lead to the following conclusion: after applying Haar transform combined with the structure-aware grouping strategy proposed by HBLLM, there is a positive correlation between the improvement in data concentration and the enhancement of model fidelity after quantization.

## C.4    Empirical Analysis of Saliency Ratio Distribution Across Layers and Blocks

To better understand the structure-aware quantization behavior of our method, we analyze how saliency ratio is distributed across different types of layers and transformer blocks. A saliency ratio is defined as the proportion of weights selected by our Hessian-based criterion during quantization.

We visualize the results using two plots: a histogram showing the saliency ratio distribution across different layer types, and a line plot depicting the evolution of block-wise saliency ratio across the network. As shown in Figure C.3a and Figure C.3b, query and key projections exhibit low saliency (mostly below $5\%$), while value, gate, and up-projection layers tend to have significantly higher saliency. Additionally, we observe a gradual increase in saliency ratio from shallow to mid transformer blocks, followed by stabilization.

These results confirm that saliency ratio is not uniformly distributed, but highly dependent on layer type and block depth. This validates our strategy of adapting quantization granularity according to the structure of the model.

# D    Storage Cost Analysis and Inference Execution Process

To comprehensively assess compression effectiveness, we divide the stored data into three types: weight overhead , coefficient overhead, and flag overhead.

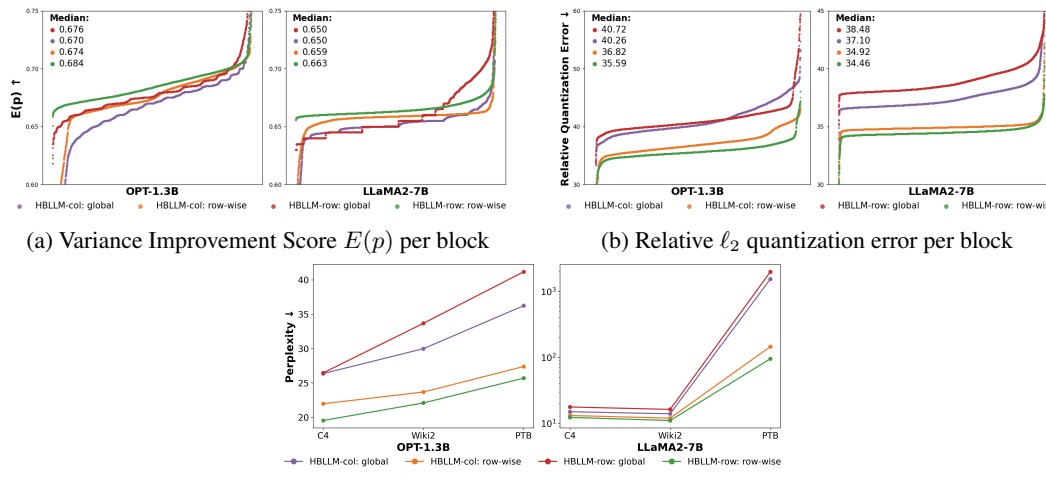

(a) Variance Improvement Score $E(p)$ per block

(b) Relative $\ell_2$ quantization error per block

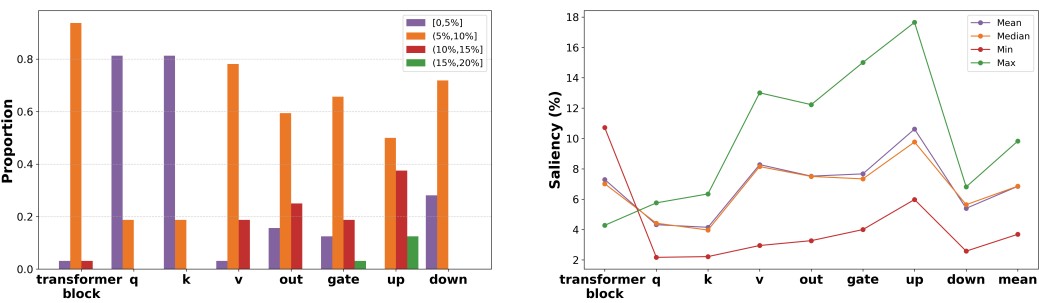

(c) Perplexity Comparison

Figure C.2: Comparative Evaluation of Grouping Strategies on Data Concentration, Quantization Error, and Model Perplexity.

(a) Saliency Ratio Distribution Across Layers and Blocks

(b) Saliency Statistics Across Layers and Blocks

Figure C.3: The Distribution of Saliency between Different Types of Weight Layers and across Different Transformer Blocks.

**Weight Overhead (W-bits)**. This refers to the number of bits used to store binarized weights. In standard 1-bit schemes, each weight requires only 1 bit. However, in schemes that retain salient weights, these may be stored with higher precision (e.g., 2-bit or 8-bit), increasing the overall weight overhead.

To increase fidelity, some weights would use more than 1 bit. For example, the salient part employs two 1-bit values with residual approximation. As a result, the average weight overhead per weight (denoted as W-bits) of Billm results would become fractional, such as 1.08 bits.

**Coefficient Overhead (C-bits)**. This refers to the additional bit-width required to store scaling factors and means. For example, OneBit introduces two scaling vectors per row or column, while ARB-LLM$_{RC}$ further computes scaling factors for both rows and columns. Although these parameters are smaller in size than the weight matrix, they must be tightly controlled in precision-critical applications.

**Flag Overhead (F-bits)**. These bits are used to store indicators for salient/non-salient weights (such as the "salient column" tag in PB-LLM or bitmap/group masks in BiLLM), or group affiliation information (e.g., group IDs).

To objectively evaluate the compression efficiency of various methods, we use the "average bit-width per weight" (**Average-Bit**) as a unified metric. This metric characterizes not only storage cost but also the bandwidth required from memory to GPU registers. The average bit-width is computed as:

$$\text{AvgBit} = \frac{\text{Total storage bits}}{\text{Total number of parameters}} \times \text{Structure expansion factor}. \tag{33}$$

here, the total number of parameters refers to the product of the matrix dimensions. The structure expansion factor accounts for mismatches in the number of stored units and original parameters (e.g., 1 for non-restructuring methods, $> 1$ for methods like FrameQuant).

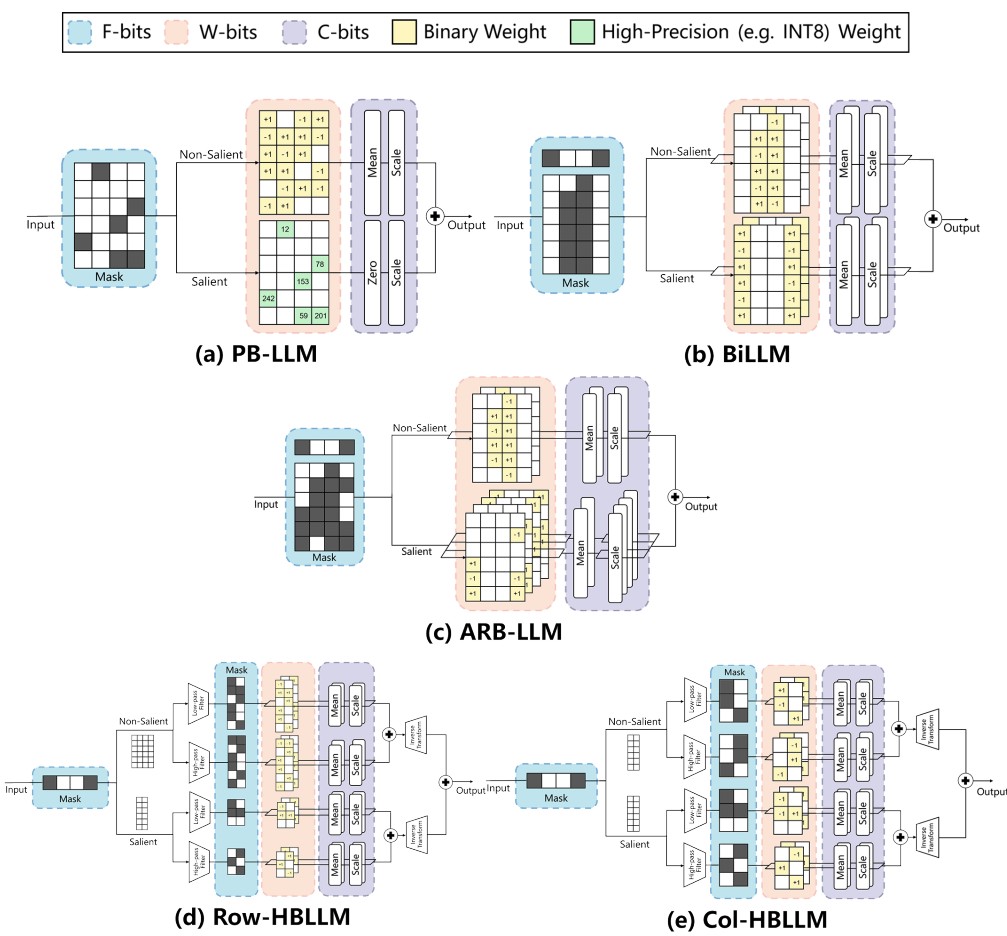

Figure D.1: Overview of storage and inference procedure across different LLM binarization methods.

Table D.1: Storage data composition of different LLM binarization methods.

| Method | W-Bits | F-Bits | C-Bits | Average-Bits |
|---|---|---|---|---|
| PB-LLM | 1.70 | 1.000 | 0.500 | 3.200 |
| BiLLM | 1.09 | 1.008 | 0.875 | 2.973 |
| ARB-LLM | 1.09 | 1.008 | 1.25 | 3.348 |
| HBLLM-row | 1.08 | 1.088 | 1.25 | 3.418 |
| HBLLM-col | 1.00 | 1.008 | 0.875 | 2.883 |

## D.1 Data Distributions of Various Binarization Methods

Taking LLaMA1-7B as an example, Figure D.1 and Table D.1 illustrate how different LLM binarization methods distribute their storage data. PB-LLM uses a hybrid-precision quantization strategy that encodes $10\%$ of weights with 8-bit asymmetric linear quantization and binarizes the remaining $90\%$. A 1-bit flag differentiates salient from non-salient weights. Non-salient parts are recovered using shared scaling and mean vectors, while salient weights use separate parameters and zero-points. All coefficients are stored in FP16 and shared per output channel, resulting in 0.5 bits of coefficient overhead per weight. PB-LLM's AvgBit is 3.2 bits.

BiLLM extends basic binarization with residual approximation for salient weights. Columns are divided into salient and non-salient parts. $90\%$ of weights use standard 1-bit encoding, while the salient portion is enhanced with residual binarization. This yields a weight overhead of 1.09 bits per weight. Additional structure information (bitmap, grouping) accounts for 1.008 bits per weight. With FP16-stored coefficients and two sets of residual parameters for salient parts, total coefficient overhead is 0.875 bit per weight. Hence, BiLLM's AvgBit is 2.973 bits.

ARB-LLM introduces group modeling for salient columns based on BiLLM to better capture complex distributions. weight overhead remains 1.09 bits per weight. Structure metadata (CGB) takes 1.008 bits per weight. All

coefficients are stored in FP16. Salient columns are grouped and assigned two sets of second-order coefficients, bringing coefficient overhead to 1.25 bits per weight. ARB-LLM's AvgBit reaches 3.348 bits.

HBLLM extends BiLLM by introducing structure-aware grouping strategies to improve quantization fidelity in the Haar domain. The weight matrix is split into salient and non-salient parts. Salient columns undergo column-wise Haar transforms, while non-salient parts are transformed along either row or column directions, followed by grouped binarization. All weights are quantized using standard 1-bit encoding. HBLLM-col has a weight overhead of 1.00 bit per weight; HBLLM-row employs a neighborhood averaging strategy (FillAvg) to reconstruct missing values, increasing the weight overhead to 1.08 bits/weight.Saliency bitmaps and frequency-aware grouping metadata add 1.008 bits per weight. All reconstruction coefficients are stored in FP16. HBLLM-row forms four subgroups per row with independent scale and mean values, resulting in a coefficient overhead of 1.25 bits per weight. HBLLM-col shares four subgroups across two rows, averaging two groups per row, and applies intra-band mean sharing to reduce coefficient overhead to 0.875 bit per weight. The final average bit-widths of HBLLM-row and HBLLM-col are 3.418 and 2.883 bits, respectively.

## D.2 Details of Average Bit-Width Calculation

The average bit-width of a quantized matrix $\widehat{\mathbf{W}} \in \mathbb{R}^{n \times m}$ is defined as the total memory cost (in bits) divided by the number of elements in the original matrix:

$$\textbf{AvgBit} = \frac{\mathcal{M}}{n \times m}. \tag{34}$$

For $\mathbf{W} \in \mathbb{R}^{n \times m}$, block size $k$, the memory of $\widehat{\mathbf{W}}$ after standard row-wise binarization is

$$\mathcal{M}^{\text{1st}} = \overbrace{n \times m}^{\mathbf{B}} + \overbrace{\lceil m/k \rceil}^{\text{multiple blocks}} \times \overbrace{2 \times n \times 16}^{\text{row-wise FP16 } \alpha \text{ and } \mu}. \tag{35}$$

Moreover, second-order row-wise binarization can be represented as

$$\mathcal{M}^{\text{2nd}} = \overbrace{2 \times n \times m}^{\mathbf{B}_1 \text{ and } \mathbf{B}_2} + \overbrace{\lceil m/k \rceil}^{\text{multiple blocks}} \times \overbrace{3 \times n \times 16}^{\text{row-wise FP16 } \alpha_1, \alpha_2, \text{ and } \mu}, \tag{36}$$

since row-wise $\mu_1$ and $\mu_2$ can be combined together as $\mu = \mu_1 + \mu_2$. Thus, the memory required by BiLLM can be formulated as

$$\mathcal{M}_{\text{BiLLM}} = \overbrace{2 \times n \times c + \lceil m/k \rceil \times 3n \times 16}^{\text{second-order binarization}} \tag{37}$$

$$+ n \times (m - c) + \underbrace{\overbrace{\lceil m/k \rceil \times 2n \times 16 \times 2}^{\text{first-order binarization}}}_{\text{2 groups}} + \overbrace{n \times m}^{\text{group bitmap}} + \overbrace{\widetilde{m}}^{\text{salient column bitmap}}, \tag{38}$$

where $c$ is the number of salient columns for $\mathbf{W}$.

Similarly, we can formulate the memory occupation of first-order row-column-wise binarization and ARB-RC as

$$\mathcal{M}_{\text{ARB-RC}} = \overbrace{2 \times n \times c + \underbrace{(\lceil m/k \rceil \times 2n + 2c) \times 16}_{\text{2 groups}}}^{\text{second-order binarization}} \tag{39}$$

$$+ n \times (m - c) + \underbrace{\overbrace{(\lceil m/k \rceil \times n + (m - c)) \times 16 \times 2}^{\text{first-order binarization}}}_{\text{2 groups}} + \overbrace{n \times m}^{\text{group bitmap}} + \overbrace{\widetilde{m}}^{\text{salient column bitmap}}. \tag{40}$$

In addition, since CGB is used in the experiments, the total memory of ARC-RC + CGB is

$$\mathcal{M}_{\text{ARB-RC+CGB}} = \overbrace{2 \times n \times c + \underbrace{(\lceil m/k \rceil \times 2n + 2c) \times 16 \times 2}_{\text{2 groups}}}^{\text{second-order binarization}} \tag{41}$$

$$+ n \times (m - c) + \underbrace{\overbrace{(\lceil m/k \rceil \times n + (m - c)) \times 16 \times 2}^{\text{first-order binarization}}}_{\text{2 groups}} + \overbrace{n \times m}^{\text{group bitmap}} + \overbrace{\widetilde{m}}^{\text{salient column bitmap}}. \tag{42}$$

Furthermore, we formulate the memory cost of PBLLM by considering both unsalient weights and salient weights as

$$\mathcal{M}_{\text{PBLLM}} = \overbrace{r_{\text{binary}} \times n \times m + \lceil m/k \rceil \times 2n \times 16}^{\text{unsalient weights}} \tag{43}$$

$$+ \overbrace{(1 - r_{\text{binary}}) \times n \times m \times 8 + \lceil m/k \rceil \times 2n \times 16}^{\text{salient weights}} + \overbrace{n \times m}^{\text{group bitmap}}, \tag{44}$$

where $r_{\text{binary}}$ denotes the ratio of the binarized weights.

HBLLM-row adopts four subgroups per row with independent $\alpha$ and $\mu$, intra-band mean sharing, and a neighborhood-based reconstruction strategy (FillAvg), which increases the group bitmap cost.

$$\mathcal{M}_{\text{HBLLM-row}} = \overbrace{n \times m + \underbrace{\lceil m/k \rceil \times 3n \times 16 \times 2}_{\text{2 groups}}}^{\text{unsalient weights}} \tag{45}$$

$$+ \overbrace{n \times c + \underbrace{\lceil m/k \rceil \times 2n \times 16 \times 2}_{\text{2 groups}}}^{\text{salient weights}} + \overbrace{n \times (m + c)}^{\text{group bitmap}} + \overbrace{\widetilde{m}}^{\text{salient column bitmap}}. \tag{46}$$

HBLLM-col shares four subgroups across two rows and applies intra-band mean sharing.

$$\mathcal{M}_{\text{HBLLM-col}} = \overbrace{n \times (m - c) + \underbrace{\lceil m/k \rceil \times 1.5n \times 16 \times 2}_{\text{2 groups}}}^{\text{unsalient weights}} \tag{47}$$

$$+ \overbrace{n \times c + \underbrace{\lceil m/k \rceil \times 2n \times 16 \times 2}_{\text{2 groups}}}^{\text{salient weights}} + \overbrace{n \times m}^{\text{group bitmap}} + \overbrace{\widetilde{m}}^{\text{salient column bitmap}}. \tag{48}$$

**Example: Average Bit-width of HBLLM-row and HBLLM-col**

Assume $\mathbf{W} \in \mathbb{R}^{n \times m}$, block size $k = 128$ and the number of salient columns is $c = 0.08m$. The total memory cost for HBLLM-row can be expressed as:

$$\mathcal{M}_{\text{HBLLM-row}} = nm + \left\lceil \frac{m}{k} \right\rceil \cdot 3n \cdot 16 \cdot 2 + nc + \left\lceil \frac{m}{k} \right\rceil \cdot 2n \cdot 16 \cdot 2 + n(m + c) + \widetilde{m} \tag{49}$$

$$= 2nm + 2nc + 160n \left\lceil \frac{m}{k} \right\rceil + \widetilde{m}. \tag{50}$$

Thus, the average bit-width is:

$$\textbf{AvgBit} = \frac{\mathcal{M}_{\text{HBLLM-row}}}{nm} = 2.16 + \frac{160n \left\lceil \frac{m}{k} \right\rceil}{nm} + \frac{\widetilde{m}}{nm} \approx 3.418 \, \text{bits}. \tag{51}$$

Similarly, for HBLLM-col:

$$\mathcal{M}_{\text{HBLLM-col}} = n(m - c) + \left\lceil \frac{m}{k} \right\rceil \cdot 1.5n \cdot 16 \cdot 2 + nc + \left\lceil \frac{m}{k} \right\rceil \cdot 2n \cdot 16 \cdot 2 + nm + \widetilde{m} \tag{52}$$

$$= 2nm + 112n \left\lceil \frac{m}{k} \right\rceil + \widetilde{m}. \tag{53}$$

Then, the average bit-width becomes:

$$\textbf{AvgBit} = \frac{\mathcal{M}_{\text{HBLLM-col}}}{nm} = 2 + \frac{112n \left\lceil \frac{m}{k} \right\rceil}{nm} + \frac{\widetilde{m}}{nm} \approx 2.883 \, \text{bits}. \tag{54}$$

# E  HBLLM Implementation

The implementation details of the HBLLM quantization pipeline are provided in Algorithm E.1.

**Algorithm E.1** HBLLM: Detailed functions process

**func** SALIENT($\boldsymbol{W}, \boldsymbol{H}^c$)

1: $\boldsymbol{S} \leftarrow \boldsymbol{W}^2/[\boldsymbol{H}^c_{b:b+\beta,b:b+\beta}]^2$    // salient matrix
2: rows$\{\cdot\} \leftarrow \text{topk}(\|\boldsymbol{S}\|_2, \dim = 0)$
3: $e \leftarrow \infty$    // searching error
4: $K \leftarrow 0$    // optimal number of salient columns
5: **for** $i = 1, 2, \ldots, \text{len}(\text{rows})$ **do**
6: $\quad \boldsymbol{B}_1 \leftarrow \text{BINARY}(\boldsymbol{W}_{:,j \in \text{rows}[:i]})$
7: $\quad \boldsymbol{B}_2 \leftarrow \text{BINARY}(\boldsymbol{W}_{:,j \notin \text{rows}[:i]})$
8: $\quad$ **if** $\|\boldsymbol{W} - (\boldsymbol{B}_1 \cup \boldsymbol{B}_2)\|^2 < e$ **then**
9: $\quad\quad e \leftarrow \|\boldsymbol{W} - (\boldsymbol{B}_1 \cup \boldsymbol{B}_2)\|^2$
10: $\quad\quad K \leftarrow i$
11: $\quad$ **end if**
12: **end for**
13: **return** rows$[: K]$

**func** HAARQUANT($\boldsymbol{W}, \text{mode} \in \{\text{COL}, \text{ROW}\}$)

1: $\boldsymbol{W}_{\text{low}} \leftarrow \mathcal{H}_{\text{low}}(\boldsymbol{W})$
2: $p_1^* \leftarrow \text{SEG\_ROW\_SEARCH}(\boldsymbol{W}_{\text{low}})$
3: $\boldsymbol{B}_{\text{low}} \leftarrow \text{BINARY}(\boldsymbol{W}^{(b)}_{|w_{i,j}| \leq p_1^*, \text{low}}) +$
$\quad \text{BINARY}(\boldsymbol{W}^{(b)}_{|w_{i,j}| > p_1^*, \text{low}})$
4: $\boldsymbol{W}_{\text{diff}} \leftarrow \boldsymbol{W} - \mathcal{H}^{-1}(\boldsymbol{B}_{\text{low}}, \boldsymbol{0})$
5: $\boldsymbol{W}_{\text{high}} \leftarrow \mathcal{H}_{\text{high}}(\boldsymbol{W}_{\text{diff}})$
6: $p_2^* \leftarrow \text{SEG\_ROW\_SEARCH}(\boldsymbol{W}_{\text{high}})$
7: $\boldsymbol{B}_{\text{high}} \leftarrow \text{BINARY}(\boldsymbol{W}^{(b)}_{|w_{i,j}| \leq p_1^*, \text{high}}) +$
$\quad \text{BINARY}(\boldsymbol{W}^{(b)}_{|w_{i,j}| > p_1^*, \text{high}})$
8: $\boldsymbol{B} \leftarrow \boldsymbol{B}_{\text{low}} + \boldsymbol{B}_{\text{high}}$
9: **return** $\boldsymbol{B}$

**func** BINARY($\boldsymbol{W}$)

1: $\mu \leftarrow \frac{1}{m} \sum_{j=1}^m W_{.j}$    // row-wise mean
2: $\widetilde{\boldsymbol{W}} \leftarrow \boldsymbol{W} - \mu$    // centered matrix
3: $\alpha = \sqrt{\frac{\|\widetilde{\boldsymbol{W}}\|_2^2}{m}}$    // row-wise scale
4: $\boldsymbol{B} \leftarrow \alpha \cdot \text{sign}(\widetilde{\boldsymbol{W}}) + \mu$
5: **return** $\boldsymbol{B}$

**func** SEG\_ROW\_SEARCH($\boldsymbol{W}$)

1: $n \leftarrow$ number of rows in $\boldsymbol{W}$
2: $\boldsymbol{e} \leftarrow +\infty \times \mathbf{1}_{n \times 1}$    // row-wise error
3: $\boldsymbol{p}^* \leftarrow \mathbf{0}_{n \times 1}$
$\quad$ // optimal break-point of each row
4: **for** $\tau = 0.1, 0.2, \ldots, 0.9$ **do**
5: $\quad \boldsymbol{p} \leftarrow \tau \times \max(\text{abs}(\boldsymbol{W}).(\dim = 1))$
6: $\quad \boldsymbol{B}_1 \leftarrow \text{BINARY}(\boldsymbol{W}_{|w_{i \in [n],:}| \leq \boldsymbol{p}})$
7: $\quad \boldsymbol{B}_2 \leftarrow \text{BINARY}(\boldsymbol{W}_{|w_{i \in [n],:}| > \boldsymbol{p}})$
8: $\quad$ **if** $\|\boldsymbol{W} - (\boldsymbol{B}_1 + \boldsymbol{B}_2)\|^2 < e$ **then**
9: $\quad\quad e \leftarrow \|\boldsymbol{W} - (\boldsymbol{B}_1 + \boldsymbol{B}_2)\|^2$
10: $\quad\quad \boldsymbol{p}^* \leftarrow \boldsymbol{p}$
11: $\quad$ **end if**
12: **end for**
13: **return** $\boldsymbol{p}^*$

Table F.1: Perplexity and zero-shot accuracy results of DeepSeek-R1-Distill-Llama-8B.

| Model | Method | Wbits | Perplexity↓ | | | AvgQA↑ |
|---|---|---|---|---|---|---|
| | | | C4 | Wiki2 | PTB | |
| | FullPrecision | 16.00 | 18.40 | 13.14 | 20.57 | 63.80 |
| DeepSeek-R1-Distill-Llama-8B | FrameQuant | 2.20 | 59.60 | 46.71 | 70.79 | 43.95 |
| | PB-LLM | 1.70 | 316.3 | 224.6 | 344.3 | 34.52 |
| | BiLLM | 1.06 | 234.4 | 219.5 | 442.9 | 35.91 |
| | ARB-LLM$_X$ | 1.06 | 74.31 | 54.73 | 69.35 | 42.92 |
| | ARB-LLM$_{RC}$ | 1.06 | 67.77 | 54.27 | 92.37 | 43.00 |
| | **HBLLM-row** | 1.05 | **40.88** | **29.26** | **45.00** | **47.06** |
| | **HBLLM-col** | 1.00 | 55.82 | 35.80 | 62.80 | 45.71 |

# F  Additional Experimental Results

## F.1  Experimental Results for DeepSeek-R1-Distill-Llama-8B

Table F.1 provides further experimental results on DeepSeek-R1-Distill-Llama-8B model. In line with the trends observed in Table 1, HBLLM consistently outperforms existing 1-2 bit quantization techniques across all these evaluation metrics. Remarkably, even when applied to the more complex and modern DeepSeek-R1-Distill-Llama-8B architecture, HBLLM preserves the same performance advantages previously demonstrated on LLaMA3-8B. This assessment highlights the effectiveness of HBLLM for LLMs.

Table F.2: Accuracy of 9 QA datasets on LLaMA1 family models. We compare the results among FrameQuant, PB-LLM, BiLLM, ARB-LLM and HBLLM to validate the quantization effect.

| LLaMA1 | | | Zero-shot Accuracy↑ | | | | | | | | | AvgQA↑ |
|---|---|---|---|---|---|---|---|---|---|---|---|---|
| Size | Method | Wbits | PIQA | BoolQ | OBQA | WinoG | ARC-e | ARC-c | HSwag | COPA | LAMBD | |
| | FullPrecision | 16.00 | 78.67 | 75.02 | 34.20 | 70.01 | 75.34 | 41.89 | 56.94 | 85.00 | 73.51 | 65.62 |
| 7B | FrameQuant | 2.20 | 71.16 | 69.69 | 23.60 | **66.54** | 61.57 | 29.78 | 43.67 | **81.00** | 58.70 | 56.19 |
| | PB-LLM | 1.70 | 54.62 | 58.04 | 13.20 | 49.17 | 29.38 | 21.25 | 27.61 | 61.00 | 7.10 | 35.71 |
| | BiLLM | 1.09 | 59.63 | 54.62 | 15.00 | 53.20 | 35.27 | 19.80 | 30.38 | 71.00 | 21.15 | 40.01 |
| | ARB-LLM$_X$ | 1.09 | 63.55 | 64.10 | 16.80 | 56.75 | 42.21 | 21.93 | 34.21 | 73.00 | 38.27 | 45.65 |
| | ARB-LLM$_{RC}$ | 1.09 | 69.15 | 64.10 | 22.40 | 61.25 | 52.90 | 25.26 | 39.26 | 78.00 | 57.71 | 52.23 |
| | HBLLM-row | 1.08 | **73.67** | **71.07** | **24.60** | 62.75 | **64.81** | **30.80** | **47.60** | 79.00 | **63.03** | **57.48** |
| | HBLLM-col | 1.00 | 71.93 | 62.60 | 22.00 | 62.27 | 61.15 | 28.58 | 44.68 | 78.00 | 55.09 | 54.03 |
| | FullPrecision | 16.00 | 79.16 | 77.92 | 33.20 | 72.61 | 77.36 | 46.42 | 59.93 | 90.00 | 76.19 | 68.09 |
| 13B | FrameQuant | 2.20 | 75.14 | **73.64** | 26.20 | 68.67 | 66.71 | 34.64 | 48.58 | 83.00 | 69.59 | 60.69 |
| | PB-LLM | 1.70 | 58.00 | 62.02 | 14.60 | 52.96 | 33.25 | 18.26 | 29.99 | 69.00 | 25.44 | 40.39 |
| | BiLLM | 1.10 | 67.19 | 67.09 | 19.00 | 60.69 | 53.66 | 25.34 | 39.90 | 74.00 | 51.12 | 50.89 |
| | ARB-LLM$_X$ | 1.10 | N/A | N/A | N/A | N/A | N/A | N/A | N/A | N/A | N/A | N/A |
| | ARB-LLM$_{RC}$ | 1.10 | 72.03 | 72.51 | 24.40 | 68.11 | 64.73 | 31.14 | 45.54 | **88.00** | 69.80 | 59.58 |
| | HBLLM-row | 1.09 | **76.28** | 68.93 | **28.60** | 68.90 | **70.37** | **38.31** | **52.54** | 86.00 | **73.22** | **62.57** |
| | HBLLM-col | 1.00 | 75.03 | 70.46 | 24.80 | **69.69** | 69.02 | 34.39 | 51.01 | 87.00 | 69.82 | 61.25 |
| | FullPrecision | 16.00 | 80.96 | 82.69 | 36.00 | 75.93 | 80.35 | 52.73 | 63.35 | 90.00 | 77.55 | 71.06 |
| 30B | FrameQuant | 2.20 | 76.39 | 73.58 | 31.20 | 72.85 | 72.18 | 38.14 | 53.97 | **92.00** | 75.84 | 65.13 |
| | PB-LLM | 1.70 | 63.60 | 64.34 | 16.00 | 60.30 | 44.78 | 20.90 | 34.80 | 74.00 | 46.30 | 47.22 |
| | BiLLM | 1.11 | 71.49 | 68.87 | 24.40 | 67.96 | 63.59 | 29.69 | 44.51 | 84.00 | 68.10 | 58.07 |
| | ARB-LLM$_X$ | 1.11 | N/A | N/A | N/A | N/A | N/A | N/A | N/A | N/A | N/A | N/A |
| | ARB-LLM$_{RC}$ | 1.11 | 74.86 | 76.64 | 29.60 | **73.64** | 74.62 | 37.46 | 50.29 | 91.00 | **76.31** | 64.49 |
| | HBLLM-row | 1.10 | **77.09** | **79.72** | **32.20** | 71.67 | **74.24** | **43.34** | **55.79** | 91.00 | 75.80 | **66.76** |
| | HBLLM-col | 1.00 | 76.66 | 72.23 | 29.80 | 71.35 | 73.91 | 39.68 | 53.81 | 90.00 | 76.29 | 64.86 |
| | FullPrecision | 16.00 | 81.34 | 84.86 | 38.00 | 77.43 | 81.31 | 52.82 | 64.56 | 91.00 | 79.12 | 72.27 |
| 65B | FrameQuant | 2.20 | **79.00** | 83.27 | 32.00 | 74.66 | **77.23** | 45.56 | 56.58 | 90.00 | 78.92 | 68.58 |
| | PB-LLM | 1.70 | 71.98 | 79.45 | 28.20 | 74.98 | 67.34 | 34.47 | 46.57 | 89.00 | 70.37 | 62.48 |
| | BiLLM | 1.10 | 74.05 | 80.40 | 26.20 | 70.40 | 69.32 | 36.60 | 48.15 | 85.00 | 68.35 | 62.05 |
| | ARB-LLM$_X$ | 1.10 | N/A | N/A | N/A | N/A | N/A | N/A | N/A | N/A | N/A | N/A |
| | ARB-LLM$_{RC}$ | 1.10 | 77.97 | **83.94** | 32.60 | **75.93** | 76.64 | 44.28 | 55.58 | 90.00 | **79.84** | 68.53 |
| | HBLLM-row | 1.09 | 78.89 | 81.96 | **33.80** | 75.77 | 76.94 | **45.65** | **59.03** | 91.00 | 79.58 | **69.18** |
| | HBLLM-col | 1.00 | **79.00** | 81.47 | 31.40 | 74.51 | 75.17 | 42.66 | 57.98 | **91.00** | 77.28 | 67.83 |

## F.2 Comparison on 9 zero-shot QA datasets

In the main paper, we report the average accuracy (*AvgQA*) across 9 zero-shot QA datasets to provide a high-level comparison of different quantization methods. In this appendix, we present the detailed accuracy on each individual dataset. As shown in Tables F.2–F.5 our method consistently delivers strong performance across all datasets, further validating its effectiveness and robustness in diverse zero-shot question answering tasks.

## F.3 Comparison of Avg. Relative Perplexity and Avg. Relative QA Accuracy across Models

To provide a unified view of model performance across both language modeling and common sense reasoning tasks, we compute two metrics: **Avg. Relative Perplexity** and **Avg. Relative QA**. **Avg. Relative Perplexity** is calculated as the average over three language modeling datasets: C4, Wiki2 and PTB. **Avg. Relative QA** is computed as the average across 9 zero-shot QA datasets.

**Relative Perplexity** is defined as:

$$RS_{\text{Perplexity}} := \frac{S_{\text{Perplexity}}}{S_{\text{Perplexity}}^{\text{FP}}}, \tag{55}$$

where $S_{\text{Perplexity}}$ and $S_{\text{Perplexity}}^{\text{FP}}$ are the perplexities of the models after and before quantization, respectively.

**Relative QA** is defined as:

$$RS_{\text{QA}} := \frac{S_{\text{QA}}}{S_{\text{QA}}^{\text{FP}}}. \tag{56}$$

where $S_{\text{QA}}$ and $S_{\text{QA}}^{\text{FP}}$ are the QA scores of the models after and before quantization, respectively.

Figure F.1 presents the comparison of these two metrics across the LLaMA-1, LLaMA-2, LLaMA-3, and OPT family models. HBLLM consistently shows lower relative perplexity and higher relative QA accuracy compared to prior 1–2 bit quantization methods (PB-LLM, BiLLM, and ARB-LLM$_{RC}$), demonstrating both effectiveness and scalability across diverse LLMs.

Table F.3: Accuracy of 9 QA datasets on LLaMA2 family models. We compare the results among FrameQuant, PB-LLM, BiLLM, ARB-LLM and HBLLM to validate the quantization effect.

| LLaMA2 | | | Zero-shot Accuracy↑ | | | | | | | | | AvgQA↑ |
|---|---|---|---|---|---|---|---|---|---|---|---|---|
| Size | Method | Wbits | PIQA | BoolQ | OBQA | WinoG | ARC-e | ARC-c | HSwag | COPA | LAMBD | |
| | FullPrecision | 16.00 | 76.44 | 79.72 | 33.40 | 66.46 | 73.91 | 44.20 | 57.80 | 87.00 | 70.95 | 65.54 |
| 7B | FrameQuant | 2.20 | 68.55 | 67.13 | 19.00 | **61.80** | 56.61 | 27.56 | 42.20 | 78.00 | 53.89 | 52.75 |
| | PB-LLM | 1.70 | 54.84 | 61.90 | 11.80 | 48.07 | 27.90 | 19.45 | 27.78 | 60.00 | 17.14 | 36.54 |
| | BiLLM | 1.08 | 59.85 | 63.82 | 12.80 | 54.14 | 40.15 | 21.67 | 31.66 | 64.00 | 30.93 | 42.11 |
| | ARB-LLM$_X$ | 1.08 | 61.97 | 66.42 | 16.20 | 57.70 | 43.27 | 22.78 | 33.81 | 68.00 | 38.54 | 45.41 |
| | ARB-LLM$_{RC}$ | 1.08 | 61.15 | 59.51 | 18.00 | 59.27 | 41.75 | 23.46 | 39.62 | 67.00 | 50.61 | 46.71 |
| | HBLLM-row | 1.07 | **72.96** | **73.58** | **25.60** | 60.93 | **61.87** | **32.17** | **47.50** | **84.00** | **61.03** | **57.74** |
| | HBLLM-col | 1.00 | 71.11 | 69.66 | 21.40 | 61.33 | 59.22 | 29.35 | 45.33 | 74.00 | 55.44 | 54.09 |
| | FullPrecision | 16.00 | 79.05 | 80.55 | 35.20 | 72.14 | 79.42 | 48.46 | 60.05 | 91.00 | 76.73 | 69.18 |
| 13B | FrameQuant | 2.20 | 73.83 | 76.30 | 25.60 | **69.61** | 69.02 | 33.79 | 47.15 | 85.00 | 71.84 | 61.35 |
| | PB-LLM | 1.70 | 54.30 | 41.25 | 12.60 | 50.12 | 27.27 | 20.14 | 27.00 | 59.00 | 4.52 | 32.91 |
| | BiLLM | 1.08 | 61.81 | 66.51 | 18.00 | 56.35 | 43.64 | 21.84 | 33.33 | 75.00 | 44.34 | 46.76 |
| | ARB-LLM$_X$ | 1.08 | N/A | N/A | N/A | N/A | N/A | N/A | N/A | N/A | N/A | N/A |
| | ARB-LLM$_{RC}$ | 1.08 | 69.64 | 72.42 | 25.40 | 64.17 | 62.12 | 30.03 | 40.99 | 85.00 | 66.35 | 57.35 |
| | HBLLM-row | 1.07 | **75.63** | **77.74** | **29.80** | 68.27 | **71.21** | **37.12** | **52.54** | **88.00** | **72.19** | **63.61** |
| | HBLLM-col | 1.00 | 74.86 | 77.61 | 26.80 | 69.14 | 70.66 | 35.75 | 50.46 | 84.00 | 69.12 | 62.04 |
| | FullPrecision | 16.00 | 82.26 | 83.76 | 37.20 | 77.98 | 82.74 | 54.35 | 64.77 | 94.00 | 79.60 | 72.96 |
| 70B | FrameQuant | 2.20 | N/A | N/A | N/A | N/A | N/A | N/A | N/A | N/A | N/A | N/A |
| | PB-LLM | 1.70 | 64.04 | 74.77 | 21.20 | 64.72 | 55.64 | 27.22 | 38.77 | 80.00 | 61.94 | 54.26 |
| | BiLLM | 1.09 | 68.39 | 70.64 | 24.40 | 65.98 | 64.23 | 32.34 | 41.85 | 82.00 | 52.47 | 55.81 |
| | ARB-LLM$_X$ | 1.09 | N/A | N/A | N/A | N/A | N/A | N/A | N/A | N/A | N/A | N/A |
| | ARB-LLM$_{RC}$ | 1.09 | 77.53 | 80.21 | 34.40 | 75.85 | 77.31 | 44.80 | 55.84 | **92.00** | **80.98** | 68.77 |
| | HBLLM-row | 1.08 | **79.65** | **81.56** | **33.40** | 75.06 | **79.29** | **50.51** | 59.02 | 91.00 | 80.56 | **70.01** |
| | HBLLM-col | 1.00 | 78.84 | 78.99 | 30.80 | **76.24** | 77.99 | 45.39 | **58.61** | **92.00** | 78.59 | 68.61 |

Table F.4: Accuracy of 9 QA datasets on LLaMA3 family models. We compare the results among FrameQuant, PB-LLM, BiLLM, ARB-LLM and HBLLM to validate the quantization effect.

| LLaMA3 | | | Zero-shot Accuracy↑ | | | | | | | | | AvgQA↑ |
|---|---|---|---|---|---|---|---|---|---|---|---|---|
| Size | Method | Wbits | PIQA | BoolQ | OBQA | WinoG | ARC-e | ARC-c | HSwag | COPA | LAMBD | |
| | FullPrecision | 16.00 | 78.35 | 83.18 | 34.20 | 71.67 | 81.61 | 52.90 | 57.66 | 89.00 | 71.88 | 68.94 |
| 8B | FrameQuant | 2.20 | 65.40 | 71.87 | 19.20 | 59.04 | **60.19** | **29.69** | **47.06** | 78.00 | 46.94 | 52.27 |
| | PB-LLM | 1.70 | 57.18 | 62.26 | 11.60 | 50.36 | 31.31 | 18.00 | 28.59 | 55.00 | 17.16 | 36.83 |
| | BiLLM | 1.06 | 59.14 | 64.46 | 15.20 | 53.75 | 37.54 | 19.45 | 32.01 | 65.00 | 30.04 | 41.84 |
| | ARB-LLM$_X$ | 1.06 | 61.15 | 63.91 | 15.00 | 56.43 | 45.37 | 20.14 | 31.84 | 66.00 | 36.07 | 43.40 |
| | ARB-LLM$_{RC}$ | 1.06 | 62.73 | 69.72 | 18.80 | 56.91 | 50.34 | 25.94 | 35.18 | 74.00 | 49.08 | 49.08 |
| | HBLLM-row | 1.05 | **67.68** | **72.94** | **23.80** | **63.54** | 56.52 | 28.07 | 43.55 | **81.00** | **56.14** | **54.80** |
| | HBLLM-col | 1.00 | 67.03 | 69.94 | 21.20 | 60.93 | 49.79 | 25.85 | 42.79 | **81.00** | 44.30 | 51.43 |
| | FullPrecision | 16.00 | 82.15 | 85.38 | 37.80 | 80.51 | 86.87 | 60.07 | 66.34 | 93.00 | 79.45 | 74.62 |
| 70B | FrameQuant | 2.20 | N/A | N/A | N/A | N/A | N/A | N/A | N/A | N/A | N/A | N/A |
| | PB-LLM | 1.70 | 57.89 | 68.07 | 17.60 | 58.96 | 37.37 | 19.62 | 38.86 | 80.00 | 48.67 | 47.45 |
| | BiLLM | 1.09 | 52.39 | 42.26 | 12.60 | 51.78 | 25.34 | 20.73 | 28.11 | 65.00 | 9.41 | 34.18 |
| | ARB-LLM$_X$ | 1.09 | N/A | N/A | N/A | N/A | N/A | N/A | N/A | N/A | N/A | N/A |
| | ARB-LLM$_{RC}$ | 1.09 | **74.86** | 80.89 | 25.80 | 72.69 | **72.69** | **40.44** | 51.99 | 85.00 | 70.76 | **63.90** |
| | HBLLM-row | 1.08 | 52.88 | **83.18** | 28.80 | **78.06** | 27.19 | 20.48 | **53.83** | 89.00 | **74.64** | 56.45 |
| | HBLLM-col | 1.00 | 56.04 | 77.83 | **29.00** | 73.40 | 33.63 | 18.34 | 53.19 | **91.00** | 70.56 | 55.89 |

Table F.5: Accuracy of 9 QA datasets on DeepSeek-R1-Distill-Llama-8B. We compare the results among FrameQuant, PB-LLM, BiLLM, ARB-LLM and HBLLM to validate the quantization effect.

| DeepSeek-R1-Distill-Llama | | | Zero-shot Accuracy↑ | | | | | | | | | AvgQA↑ |
|---|---|---|---|---|---|---|---|---|---|---|---|---|
| Size | Method | Wbits | PIQA | BoolQ | OBQA | WinoG | ARC-e | ARC-c | HSwag | COPA | LAMBD | |
| | FullPrecision | 16.00 | 76.33 | 82.94 | 31.40 | 67.48 | 70.54 | 40.27 | 55.54 | 89.00 | 60.72 | 63.80 |
| 8B | FrameQuant | 2.20 | 61.48 | 65.50 | 17.00 | 55.17 | 42.05 | 22.10 | 35.42 | 67.00 | 29.81 | 43.95 |
| | PB-LLM | 1.70 | 55.01 | 59.54 | 12.60 | 48.07 | 29.08 | 16.98 | 27.56 | 52.00 | 9.82 | 34.52 |
| | BiLLM | 1.06 | 55.01 | 60.37 | 12.80 | 49.41 | 26.89 | 19.62 | 29.20 | 57.00 | 12.87 | 35.91 |
| | ARB-LLM$_X$ | 1.06 | 60.34 | 67.55 | 15.60 | 55.64 | 39.90 | 22.35 | 32.97 | 68.00 | 23.97 | 42.92 |
| | ARB-LLM$_{RC}$ | 1.06 | 60.50 | 63.15 | 15.40 | 53.51 | 40.24 | 21.76 | 34.55 | 64.00 | 33.90 | 43.00 |
| | HBLLM-row | 1.05 | 63.71 | **72.26** | 18.00 | 56.20 | 41.75 | **23.38** | 39.74 | **74.00** | **34.47** | **47.06** |
| | HBLLM-col | 1.00 | **64.53** | 68.53 | **18.40** | **56.35** | **42.42** | 22.44 | **39.75** | 70.00 | 28.99 | 45.71 |

Table F.6: Accuracy of 9 QA datasets on OPT family models. We compare the results among FrameQuant, PB-LLM, BiLLM, ARB-LLM and HBLLM to validate the quantization effect.

| OPT | | | Zero-shot Accuracy↑ | | | | | | | | | AvgQA↑ |
|---|---|---|---|---|---|---|---|---|---|---|---|---|
| Size | Method | Wbits | PIQA | BoolQ | OBQA | WinoG | ARC-e | ARC-c | HSwag | COPA | LAMBD | |
| | FullPrecision | 16.00 | 71.65 | 57.77 | 23.40 | 59.27 | 57.03 | 23.38 | 41.53 | 81.00 | 57.85 | 52.54 |
| | FrameQuant | 2.20 | 64.91 | 57.00 | 16.40 | 55.17 | 46.51 | 20.73 | 34.01 | 71.00 | 34.58 | 44.48 |
| | PB-LLM | 1.70 | 53.81 | 48.38 | 12.60 | 50.67 | 28.49 | 19.97 | 26.20 | 59.00 | 1.88 | 33.44 |
| 1.3B | BiLLM | 1.09 | 59.41 | 61.07 | 13.80 | 52.49 | 35.65 | 17.06 | 29.53 | 62.00 | 14.54 | 38.39 |
| | ARB-LLM$_X$ | 1.09 | 60.39 | 60.61 | 14.60 | 52.72 | 39.98 | 18.09 | 30.65 | 64.00 | 31.73 | 41.42 |
| | ARB-LLM$_{RC}$ | 1.09 | 65.13 | 56.88 | 17.60 | 53.75 | 47.22 | 20.05 | 33.58 | 71.00 | **42.27** | 45.28 |
| | HBLLM-row | 1.07 | **66.49** | **62.14** | 17.80 | 56.20 | **47.94** | **21.42** | **35.70** | 69.00 | 40.42 | **46.35** |
| | HBLLM-col | 1.00 | 65.61 | 50.18 | **18.20** | **56.43** | 45.75 | 20.90 | 35.25 | **72.00** | 38.02 | 44.70 |
| | FullPrecision | 16.00 | 73.83 | 60.34 | 25.00 | 61.33 | 60.73 | 26.79 | 45.88 | 77.00 | 63.63 | 54.95 |
| | FrameQuant | 2.20 | 67.36 | 61.93 | 18.20 | 57.22 | 52.23 | 22.70 | 36.88 | 75.00 | 54.71 | **49.58** |
| | PB-LLM | 1.70 | 54.79 | 62.11 | 13.00 | 50.67 | 30.13 | 19.20 | 27.35 | 69.00 | 12.30 | 37.62 |
| 2.7B | BiLLM | 1.10 | 60.45 | 62.08 | 13.20 | 53.59 | 35.90 | 20.56 | 30.54 | 63.00 | 20.88 | 40.02 |
| | ARB-LLM$_X$ | 1.10 | 63.00 | 62.35 | 15.40 | 54.78 | 42.85 | 19.45 | 32.20 | 69.00 | 42.38 | 44.60 |
| | ARB-LLM$_{RC}$ | 1.10 | 67.74 | 57.03 | 18.20 | **58.17** | 51.05 | 22.61 | 37.33 | 74.00 | **59.62** | 49.53 |
| | HBLLM-row | 1.09 | **68.66** | 58.35 | **20.20** | 56.91 | **52.86** | **22.87** | **39.49** | 72.00 | 47.86 | 48.80 |
| | HBLLM-col | 1.00 | 67.79 | **62.51** | 19.40 | 56.27 | 51.05 | 22.18 | 37.24 | **76.00** | 44.61 | 48.56 |
| | FullPrecision | 16.00 | 75.84 | 65.75 | 27.00 | 65.19 | 67.09 | 32.94 | 52.45 | 81.00 | 68.66 | 58.41 |
| | FrameQuant | 2.20 | 72.85 | 66.15 | 22.60 | 63.69 | **62.54** | 28.07 | 45.49 | 82.00 | **68.25** | 55.42 |
| | PB-LLM | 1.70 | 54.90 | 62.17 | 12.80 | 52.01 | 29.46 | 20.99 | 26.68 | 57.00 | 9.96 | 39.50 |
| 13B | BiLLM | 1.13 | 67.25 | 63.91 | 18.20 | 58.64 | 51.73 | 24.66 | 38.18 | 76.00 | 54.03 | 49.82 |
| | ARB-LLM$_X$ | 1.13 | N/A | N/A | N/A | N/A | N/A | N/A | N/A | N/A | N/A | N/A |
| | ARB-LLM$_{RC}$ | 1.13 | 73.34 | 67.92 | **23.40** | 64.25 | 60.14 | 27.30 | 44.41 | 82.00 | 67.44 | 55.35 |
| | HBLLM-row | 1.12 | **73.94** | **67.98** | 22.60 | 61.64 | 62.50 | **29.10** | **46.55** | **83.00** | 66.64 | **55.91** |
| | HBLLM-col | 1.00 | 73.56 | 66.51 | 22.00 | **64.56** | 62.04 | 28.92 | 45.65 | 82.00 | 68.06 | 55.66 |
| | FullPrecision | 16.00 | 77.58 | 70.40 | 30.20 | 68.35 | 70.03 | 34.47 | 54.28 | 82.00 | 71.47 | 62.09 |
| | FrameQuant | 2.20 | 75.08 | 70.49 | 26.60 | 64.48 | **66.79** | 30.72 | 48.37 | **82.00** | 72.04 | 59.62 |
| | PB-LLM | 1.70 | 64.47 | 62.94 | 16.40 | 53.67 | 44.65 | 21.84 | 35.95 | 70.00 | 45.37 | 46.14 |
| 30B | BiLLM | 1.13 | 71.33 | 63.91 | 21.80 | 61.80 | 57.87 | 25.09 | 41.94 | 80.00 | 64.23 | 54.22 |
| | ARB-LLM$_X$ | 1.13 | N/A | N/A | N/A | N/A | N/A | N/A | N/A | N/A | N/A | N/A |
| | ARB-LLM$_{RC}$ | 1.13 | 74.27 | 67.37 | 27.40 | 64.33 | 63.76 | 30.03 | 48.03 | 78.00 | 74.15 | 58.59 |
| | HBLLM-row | 1.12 | **75.46** | **70.80** | **26.80** | **66.22** | 66.12 | **30.97** | **49.76** | 82.00 | **72.23** | **60.04** |
| | HBLLM-col | 1.00 | 74.70 | 68.90 | 25.00 | 64.96 | 66.16 | 30.03 | 48.90 | 81.00 | 70.52 | 58.91 |

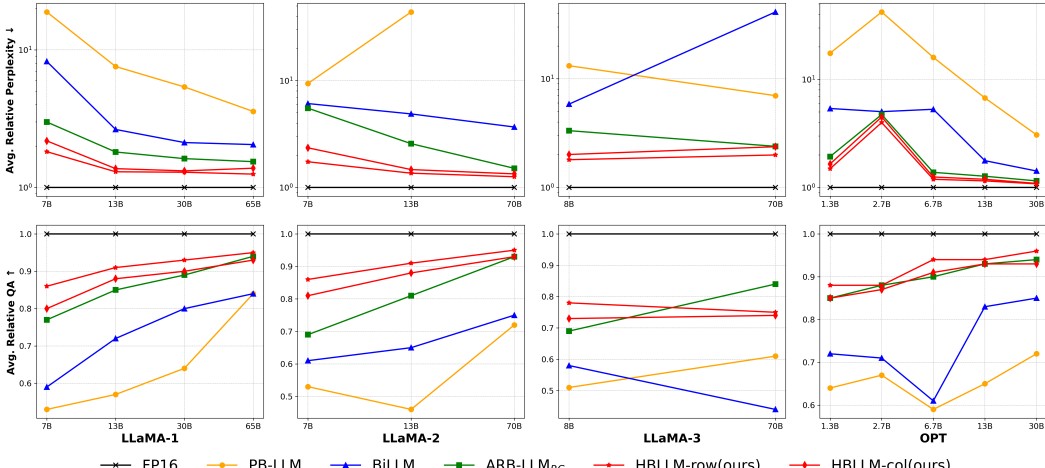

Figure F.1: Average relative perplexity and average relative QA accuracy (normalized to FP16) for LLaMA-1/2/3 family models, comparing LLM binarization methods and our HBLLM.

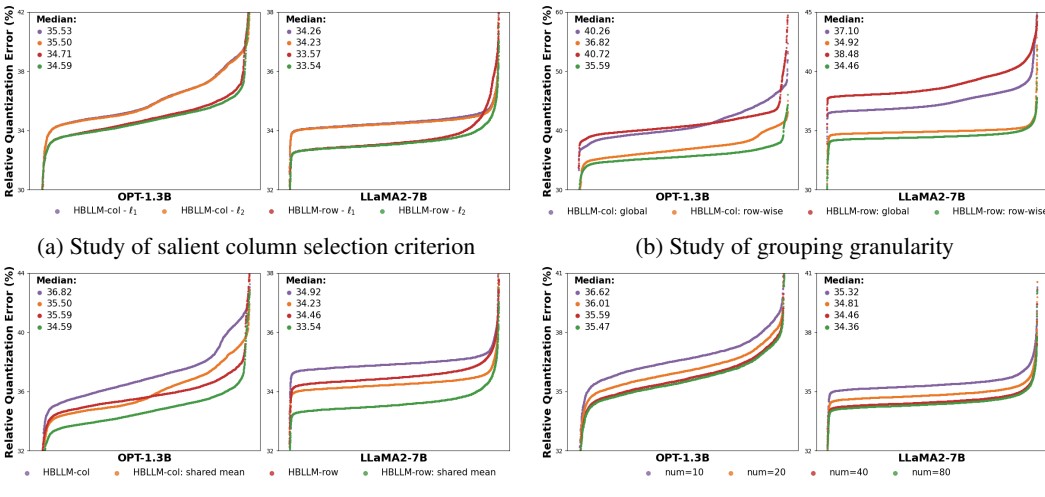

(a) Study of salient column selection criterion

(b) Study of grouping granularity

(c) Effectiveness of shared mean

(d) Study of partitioning candidates number

Figure F.2: Ablation study on OPT-1.3B and LLaMA2-7B. Results are measured by $\ell_2$ Relative Quantization Error.

## F.4 Ablation Study from the Perspective of $\ell_2$ Relative Error

To complement the ablation studies presented in the main paper, we provide an additional analysis from the perspective of relative quantization error in terms of the Frobenius norm. Specifically, we compute the relative $\ell_2$ error for each quantized weight block, defined as the Frobenius norm of the quantization residual normalized by the original weight norm. We then examine the distribution of these errors across different layers and quantization methods to assess fidelity at the matrix level.

The relative error of a quantized weight matrix is defined as:

$$RE_{\text{weight}} = \frac{\|W_{\text{FP}} - W_{\text{B}}\|_F^2}{\|W_{\text{FP}}\|_F^2}. \tag{57}$$

where $W_{\text{FP}}$ and $W_{\text{B}}$ denote the full-precision and quantized weight matrices, respectively.

Figure F.2 visualizes the sorted relative quantization errors under different ablation settings, including (a) salient column selection criterion, (b) grouping granularity, (c) shared mean strategy, and (d) number of partitioning candidates. Across all cases, our proposed HBLLM consistently achieves lower median relative error compared to prior approaches, further confirming the effectiveness of each component.

In Figure F.2a, we observe that using the $\ell_2$ norm as the column selection criterion leads to significantly lower quantization errors than $\ell_1$, validating its stronger ability to capture the energy distribution of weights. In Figure F.2b, row-wise grouping substantially outperforms global grouping, indicating the importance of fine-grained adaptivity. In Figure F.2c, the shared mean strategy slightly reduces quantization error while offering better compression efficiency. Finally, in Figure F.2d, increasing the number of partition candidates reduces error up to a point, with 40 candidates offering the best trade-off.

These observations align well with perplexity-based trends, and jointly confirm the robustness and precision of HBLLM under various quantization design choices.

## G Inference Latency Estimation and Efficiency Analysis

To evaluate the computational efficiency of HBLLM in practical inference scenarios, we design an estimation-based experiment to analyze its inference latency. This is necessary because no current inference framework fully supports the dequantization algorithm used in HBLLM, and constructing such a backend involves significant engineering effort beyond the current scope.

Our findings suggest that the inference latency of HBLLM is approximately $31.8\%$ of the FP16 baseline, indicating strong computational benefits despite the algorithm's structural complexity.

**Latency Estimation Methodology.** We define the relative inference time of HBLLM compared to FP16 as:

$$R(p, l) := \frac{T_{\text{HBLLM}}}{T_{\text{FP16}}} = (1 - p) + \frac{p}{l}, \tag{58}$$

where $p$ is the portion of time spent on matrix-vector multiplication (GEMV) during inference, and $l$ is the acceleration factor of GEMV after quantization. Following prior works such as GPTQ [**?** ], we use $p = 0.78$.

To compute $l$, we measure:

$$l := \frac{T_{\text{torch}}}{T_{\text{hqmv}}}, \tag{59}$$

where $T_{\text{torch}}$ denotes the runtime of FP16 GEMV using PyTorch, and $T_{\text{hqmv}}$ refers to our quantized matrix-vector multiplication kernel (HQMV).

Since the current hqmv implementation does not support Intra-Frequency Grouping (IFG), we use the runtime of HBLLM-col without IFG (denoted HBLLM-col w/o IFG) as a proxy to estimate the runtime of HBLLM-col with full IFG. Notably, applying IFG doubles the number of groups, potentially leading to: roughly $2\times$ more CUDA warp divergence, a $90\%$ increase in average data loading (Avgbit increases to 2.88 bit).

However, since data loading and computation can overlap on GPU, we conservatively estimate that the runtime with IFG should not exceed twice the runtime without IFG.

Table G.1: GEMV Runtime and Inference Latency Estimation for HBLLM (OPT-175B Linear Layer, P100 GPU).

| Method | $T_{\text{torch}}$ (s) | $T_{\text{hqmv}}$ (s) | $R(p, l)$ |
|---|---|---|---|
| FP16 Baseline | $1.35 \times 10^{-3}$ | — | 1.00 |
| HBLLM-col (w/o IFG) | — | $8.54 \times 10^{-5}$ | — |
| HBLLM-col (with IFG) | — | $1.70 \times 10^{-4}$ | **0.318** |

- **Setup:** We benchmark a single linear layer of OPT-175B under the same GEMV input/output shape used in GPTQ [12]. All timing experiments are conducted on an NVIDIA P100 GPU.
  - Measured: $T_{\text{torch}} = 1.35 \times 10^{-3}$s
  - Measured: $T_{\text{hqmv}}$ for HBLLM-col *w/o* IFG = $8.54 \times 10^{-5}$s
  - Estimated: $T_{\text{hqmv}}$ for full HBLLM-col (with IFG) $\approx 1.70 \times 10^{-4}$s

  This yields:

$$l = \frac{1.35 \times 10^{-3}}{1.70 \times 10^{-4}} = 7.94, \quad R = (1 - 0.78) + \frac{0.78}{7.94} \approx \boxed{0.318} \tag{60}$$

- **Discussion:** This result indicates that HBLLM inference, despite involving additional processing (e.g., dequantization and grouping), maintains high efficiency due to:
  - Lightweight binary matrix operations;
  - Reduced memory bandwidth consumption;
  - Effective overlap of memory access and computation in GPU execution.

Overall, HBLLM achieves strong acceleration over FP16 without relying on highly specialized hardware or handcrafted fusion kernels, making it a viable candidate for practical deployment in low-bit LLM inference systems.

# H   Licenses for Existing Assets

[32], llama, llama2, llama3
https://huggingface.co/TheBloke, llama, llama2, llama3
[37], https://github.com/facebookresearch/metaseq/tree/main/projects/OPT, OPT-175B LICENSE AGREEMENT
[15], https://github.com/Aaronhuang-778/BiLLM, MIT License
[18], https://github.com/ZHITENGLI/ARB-LLM, Apache License 2.0
[1], https://github.com/vsingh-group/FrameQuant
[29], https://github.com/scotfree/PbLLM, Creative Commons Zero v1.0 Universal
[4], https://leaderboard.allenai.org/physicaliqa/submissions/get-started
[7], https://github.com/google-research-datasets/boolean-questions
[8], https://huggingface.co/datasets/allenai/ai2_arc, Creative Commons Attribution Share Alike 4.0
[21], https://huggingface.co/datasets/ptb-text-only/ptb_text_only, Dataset provided for research purposes only
[22], https://huggingface.co/datasets/mindchain/wikitext2, Creative Commons Attribution-ShareAlike License.
[23], https://github.com/allenai/OpenBookQA, Apache License 2.0
[24], https://github.com/EleutherAI/lm-evaluation-harness, MIT License
[25], https://zenodo.org/records/2630551, Creative Commons Attribution 4.0 International
[26], https://github.com/google-research/text-to-text-transfer-transformer#datasets, Apache License 2.0
[27], https://asgordon.github.io/copa.html, BSD 2-Clause License

[28], https://github.com/allenai/winogrande, Apache License 2.0

