# OpenReview forum: "HBLLM: Wavelet-Enhanced High-Fidelity 1-Bit Quantization for LLMs"
_NeurIPS.cc/2025/Conference — NeurIPS 2025 spotlight_

### Official Review · Reviewer_1RBF · 2025-07-02

**Clarity:** 3
**Significance:** 2
**Originality:** 2
**Rating:** 4
**Confidence:** 4

**Summary:**

This paper proposes HBLLM, a 1-bit post-training quantization method for large language models that uses Haar wavelet transforms to improve expressiveness and accuracy. It introduces two structure-aware grouping strategies: one based on frequency and the other on saliency (L2-norm). For less important weights, it shares group means to save space. Experiments on OPT and LLaMA models show its good results, with only 1.08 bits per weight and strong performance.

**Questions:**

See weakness.

**Ethical Concerns:**

["NO or VERY MINOR ethics concerns only"]

**Final Justification:**

Some confusion has been clarified.

**Limitations:**

yes

**Quality:**

3

**Strengths And Weaknesses:**

**Strength**

1. The primary contribution is that this method is the first to integrate "localized" orthogonal transformations (i.e., Haar wavelets) into a BiLLM-style quantization process. With other 3 auxillary strategies, Frequency-aware multi-parameter intra-row grouping/L2-norm-based saliency-driven column selection/Intra-frequency-band mean sharing, is achieves comprehensive good results.

2. Experiments on comprehensive models, baselines, and evaluation benchmarks demonstrate that the proposed approach is effective.

**Weakness**

1. The main difference and contribution of this paper compared to BiLLM lies in the introduction of local orthogonal transforms, corresponding to lines 7–10 of the HBLLM algorithm. However, the actual impact of these local transforms on performance (e.g., PPL and accuracy) is not clearly demonstrated. It would be helpful to show in ablation studies what happens when this component is removed or replaced by a global orthogonal transform.

2. The authors mention in the introduction that global orthogonal transforms are computationally expensive, thus motivating the use of local transforms. However, only the overall runtime is reported. A more detailed analysis comparing the computational cost of local vs. global transforms would provide useful insight.

3. There is no evaluation or discussion of computational efficiency, such as matrix multiplication speed or end-to-end inference latency. This analysis would be important for understanding the practical benefits of the proposed method.

---

> ### Author Rebuttal · Authors · 2025-07-31
>
> Thank you very much for your careful review and constructive feedback. Here we provide point-by-point responses to your comments and questions.
>
> **W1:** The main difference and contribution of this paper compared to BiLLM lies in the introduction of local orthogonal transforms, corresponding to lines 7–10 of the HBLLM algorithm. However, the actual impact of these local transforms on performance (e.g., PPL and accuracy) is not clearly demonstrated. It would be helpful to show in ablation studies what happens when this component is removed or replaced by a global orthogonal transform.
>
> **Response:**
> Local orthogonal transforms (such as Haar) indeed significantly enhance the performance of HBLLM. Removing these transforms or replacing them with global transforms clearly leads to degraded performance. To systematically demonstrate the effectiveness of the Haar transform, we have conducted the following ablation studies and compared the results against HBLLM and BiLLM (see Table 1):
>
> 1. BiLLM+$\ell_2^\dagger$: Employing $\ell_2$-norm-based saliency-driven column selection together with multi-parameter intra-row grouping (results in Table 1 under BiLLM+$\ell_2^\dagger$).
>
> 2. Haar+BiLLM: This refers to the method obtained by removing the $\ell_2$-norm-based saliency-driven column selection and multi-parameter intra-row grouping strategies from HBLLM (see the results of Row-Haar+BiLLM and Col-Haar+BiLLM in Table 1).
>
> 3. DCT+BiLLM: Applying the BiLLM algorithm to the coefficient matrices obtained by performing 1D DCT transform row-wise or column-wise on the weight matrices (results in Table 1 under Row-DCT+BiLLM and Col-DCT+BiLLM).
>
> **Table 1: Perplexity (↓, C4, Wiki2, PTB) and AvgQA accuracy (↑, AvgQA over 9 zero-shot tasks) of BiLLM variants with Haar. OPT-1.3B (cols 2–5), LLaMA2-7B (cols 6–9).**
>
> |Method|C4↓|Wiki2↓|PTB↓|AvgQA↑|C4↓|Wiki2↓|PTB↓|AvgQA↑|
> |------|----|------|-----|-------|----|------|-----|-------|
> |BiLLM|56.24|68.43|119.2|38.39|33.97|31.38|373.0|42.11|
> | BiLLM+$\ell_2^\dagger$|56.88|70.48|92.16|39.28|28.17|25.08|226.3|41.77|
> |Row-Haar+BiLLM|47.45|52.81|62.81|39.57|25.77|25.12|138.0|44.67|
> |Col-Haar+BiLLM|95.56|128.8|171.3|36.92|41.03|37.25|5193|39.60|
> |Row-DCT+BiLLM|8010|11517|6729|31.36|45358|49395|26888|34.48|
> |Col-DCT+BiLLM|107.1|150.5|250.1|34.19|26.54|24.64|1202|44.82|
> |HBLLM-row|**19.55**|26.95|**25.70**|**46.87**|**13.00**|13.20|**85.50**|50.86|
> |HBLLM-col|21.98|**23.69**|27.39|45.21|13.18|**12.02**|146.1|**51.34**|
>
> *Notes: $\ell_2$ denotes $\ell_2$-norm-based saliency-driven column selection; $\dagger$ denotes multi-parameter intra-row grouping. In Table 1, the results of HBLLM-row and HBLLM-col (our proposed method) already include these strategies.*
>
> From Table 1, we summarize the following observations:
>
> - BiLLM+$\ell_2^\dagger$ slightly improves performance compared to the original BiLLM.
>
> - The HBLLM method demonstrates substantial performance improvement compared to BiLLM+$\ell_2^\dagger$ (see BiLLM+$\ell_2^\dagger$, HBLLM-row, and HBLLM-col rows in Table 1).
>
> - Row-Haar+BiLLM notably outperforms BiLLM and BiLLM+$\ell_2^\dagger$, yet still lags significantly behind HBLLM.
>
> - Col-Haar+BiLLM shows clearly inferior performance compared to BiLLM.
>
> - Both HBLLM-row and HBLLM-col outperform BiLLM and BiLLM+$\ell_2^\dagger$ substantially, highlighting the critical role of an effective quantization strategy that leverages the Haar transform to preserve the frequency-domain structural features of weights.
>
> - Using the global row-wise DCT leads to severe performance degradation, significantly underperforming BiLLM. While the global column-wise DCT shows some performance improvements in certain tests for LLaMA2-7B, it still significantly underperforms BiLLM for all tests with OPT-1.3B.
>
> Overall, local orthogonal transforms are indeed beneficial in preserving structural fidelity during quantization. However, replacing them with global transforms would necessitate re-designing the quantization strategy to maintain the model's performance.
>
> **W2:** The authors mention in the introduction that global orthogonal transforms are computationally expensive, thus motivating the use of local transforms. However, only the overall runtime is reported. A more detailed analysis comparing the computational cost of local vs. global transforms would provide useful insight.
>
> **Response:** Quantization algorithms leveraging local orthogonal transforms demonstrate a clear advantage in runtime efficiency. To substantiate this, we compared the runtime and computational complexity of model quantization using Haar (local transform) versus DCT (global transform), as shown in Table 2 below.
>
> **Table 2: Time comparison between BiLLM, DCT+BiLLM and HBLLM on LLaMA-1 with different model sizes.**
>
> |Method|7B|13B|30B|
> |-|-|-|-|
> |BiLLM|36min|71min|142min|
> |DCT+BiLLM|211min|414min|1012min|
> |HBLLM|44min|98min|173min|
>
> The experimental results indicate that the computational overhead introduced by global transforms is substantial. HBLLM's runtime is moderately higher ($<25$%) than BiLLM’s, whereas the DCT-based transform incurs a runtime cost approximately 5–7 times greater (using NumPy’s DCT implementation).
>
> Regarding computational complexity:
>
> - Haar (local transform) involves only pairwise operations between adjacent elements. When a fixed number of decomposition levels is used (e.g., one-level transform), it results in linear computational complexity $O(n)$, where $n$ is the number of elements in a weight matrix. Such locality enables these operations to be fused into GPU kernels for concurrent execution without introducing significant additional memory overhead.
>
> - DCT (global transform) requires computation across the entire dimension, with complexity of $O(n \log_2 n)$ using an FFT-based implementation. In contrast, applying the row-wise (or col-wise) 1D Haar transform results in a total complexity of $O(n)$. For instance, consider the fully-connected layers in LLaMA-30B, which have dimensions of $6656\times6656$. Applying a 1D DCT in this scenario increases computational complexity approximately 13-fold compared to the row-wise 1D Haar transform.
>
> In summary, both our theoretical analysis and experimental results clearly demonstrate that local orthogonal transforms (such as Haar) provide a superior balance between accuracy and efficiency, exhibiting significant practical advantages for real-world deployment.
>
>
> **W3:** There is no evaluation or discussion of computational efficiency, such as matrix multiplication speed or end-to-end inference latency. This analysis would be important for understanding the practical benefits of the proposed method.
>
> **Response:** To address this point, we designed an experiment that combines direct measurement with estimation to evaluate the inference latency of HBLLM. This is necessary because, at present, there is no existing inference framework that fully supports the dequantization algorithm used in HBLLM. Building a highly optimized inference framework requires considerable engineering effort and cannot be completed in the short term.
>
> Our estimation results show that the inference latency of HBLLM is approximately $31.8$% of the FP16 baseline inference time.
>
> **Estimation Procedure**
>
> We define the relative inference time of HBLLM to the FP16 baseline as:
> \begin{equation}
>  R(p,l):= \frac{T_{HBLLM}}{T_{fp16}}=\left[ (1-p)+p/l \right].
> \tag{1}
> \end{equation}
> where:
> - $ T_{HBLLM} = p*T_{fp16} $ corresponds to the portion of inference time attributed to General Matrix–Vector multiplication (GEMV) ($p$ is the proportion of inference time spent on GEMV, which depends on the LLM architecture. Following GPTQ’s benchmark [1], we set $p=0.78$)
> - $l$ is the GEMV acceleration factor after quantization, defined as:
>  \begin{equation}
>  l:= \frac{T_{torch}}{ T_{hqmv}}.
> \tag{2}
> \end{equation}
> where $T_{torch}$ is the runtime of FP16 GEMV using PyTorch, and $T_{hqmv}$ is the runtime of our quantized GEMV implementation (denoted hqmv), which varies with the quantization strategy.
>
> Since the current hqmv implementation does not support Intra-Frequency Grouping (IFG), we use the runtime of HBLLM-col without IFG (denoted HBLLM-col w/o IFG) as a proxy to estimate the runtime of HBLLM-col with full IFG. Notably, applying IFG doubles the number of groups, potentially leading to:
> - roughly 2× more CUDA warp divergence,
> - a $90$% increase in average data loading (Avg bits per weight increases to 2.88).
>
> However, since data loading and computation can overlap on GPU, we conservatively estimate that the runtime with IFG should not exceed twice the runtime without IFG.
>
> **Experiment Steps**
>
> The full estimation process involves the following steps:
>
> 1. GEMV Runtime Measurement: Measure $T_{torch}$ and $T_{hqmv}$ using the HBLLM-col w/o IFG strategy.
>
> 2. Runtime Estimation with IFG: Estimate the runtime of hqmv with full HBLLM-col (i.e., with IFG), and compute the acceleration factor $l$.
>
> 3. Final Latency Ratio: Plug $p=0.78$ and the estimated $l$ into Equation (1) to compute the relative inference time
> $R$.
>
> **Experimental Results**
>
> Following the GPTQ benchmark setup [1], we tested GEMV on a single linear layer from the OPT-175B model. Experiments were run on an NVIDIA P100 GPU.
>
>
> The result from a single run showed $ T_{torch} = 1.35e-03s $ , and $ T_{hqmv} $ with HBLLM-col w/o IFG was $ 8.54e-05s $.
> Then, we estimated
> $T_{hqmv}$ with HBLLM-col w/o IFG to be $1.70e-04s$. Finally, substituting $p=0.78$ and $l=7.94$ into the equation yields
> $R=0.318.$
>
> This means that the estimated inference latency of HBLLM is only $31.8$% of that of the FP16 baseline, demonstrating its significant computational efficiency.
>
> [1] Elias Frantar, Saleh Ashkboos, Torsten Hoefler, Dan Alistarh.GPTQ: Accurate Post-training Compression for Generative Pretrained Transformers. arXiv preprint arXiv:2210.1732, 2022.

---

> > ### Comment · Reviewer_1RBF · 2025-08-05
> > **Response to the Authors**
> >
> > Dear author,
> >
> > I read your reply in full and clarified some confusion. Look forward to seeing your better revised paper. Best wishes.

---

> > > ### Author Response · Authors · 2025-08-07
> > >
> > > Thank you again for reviewing our response and for your constructive feedback. We appreciate your suggestions regarding the ablation of local orthogonal transforms, the comparison with global transforms, and the analysis of computational efficiency. We will incorporate these points into the revised version and include a more detailed discussion of practical efficiency metrics such as matrix multiplication cost and inference latency.

---

### Official Review · Reviewer_cU7V · 2025-07-03

**Clarity:** 3
**Significance:** 3
**Originality:** 3
**Rating:** 5
**Confidence:** 3

**Summary:**

The paper proposes the HBLLM approach to enhance 1-bit post-training quantization for LLMs. HBLLM uses Haar wavelet transforms to decompose weights by frequency, enabling more effective quantization. The proposed method introduces structure-aware grouping and saliency-based selection to improve accuracy and storage efficiency. Results on the OPT and LLaMA models show that HBLLM achieves state-of-the-art performance for 1-bit quantization.

**Questions:**

1. What is the difference between the proposed saliency-driven selection via L2 norm and AWQ [1] ?

2. The objective function in the paper does not consider calibration data. However, the authors use calibration data to compute the parameter importance metric. What would happen if the reconstruction loss also considered the calibration data?

3. As far as I know, the frequency domain is often used to capture the characteristics of data in the feature space rather than the weight space. Please correct me if I am wrong. Why does transforming weights to the frequency domain using the Haar Transform help capture the structural differences of weights?

4. In Table 1, could you provide the equation used to compute the average bit-width in detail? Since element-wise saliency detection creates overhead, why is the bit-width using HBLLM-col 1.00?

Ref: Lin, Ji, et al. "AWQ: Activation-aware Weight Quantization for LLM Compression and Acceleration". CoRR abs/2306.00978 (2023). 2023.

**Ethical Concerns:**

["NO or VERY MINOR ethics concerns only"]

**Final Justification:**

Good paper. Most of my concerns have been resolved. Therefore, I increase the score after the rebuttal.

**Limitations:**

Yes

**Paper Formatting Concerns:**

The equations should be numbered.

**Quality:**

3

**Strengths And Weaknesses:**

Strengths:

- Good empirical improvements.
- The paper is easy to follow.
- The paper provides extensive experiments across multiple LLMs and tasks.

Weaknesses:

- The paper detects the saliency parameters in an element-wise manner, which could result in more overhead compared to a channel-wise approach. Did the authors consider this?
- The motivation for using the Haar Transform to capture the structural differences of weights is not clear.
- The paper does not number the equations.

---

> ### Author Rebuttal · Authors · 2025-07-31
>
> We sincerely thank the reviewer for the careful reading and constructive feedback. Below we provide point-by-point responses to the Weaknesses and Questions, supported by both experiments and theoretical analysis.
>
> **W1:** The paper detects the saliency parameters in an element-wise manner, which could result in more overhead compared to a channel-wise approach.
>
> **Response:** Our method does not perform element-wise saliency detection. Instead, we evaluate saliency column-wise, which incurs much lower theoretical overhead than element-wise. Specifically, as described in Algorithm E.1 of the Appendix, we first compute a saliency score for each column. Then, within each block, we perform a structured search over the sorted saliency scores to select the optimal subset of salient columns. The strategy is adapted from BiLLM [1], with the difference that we replace the original $\ell_1$-based metric with an $\ell_2$-based one.
>
> **W2:** The motivation for using the Haar Transform to capture the structural differences of weights is not clear.
>
> **Response:** Applying the Haar wavelet transform to LLM quantization offers three main advantages:
>
> * The inverse quantization set becomes richer in representation, which can be demonstrated via the CIQ metric. Please refer to the details in in the Appendix B.
>
> * The data distribution becomes more concentrated: with approximately $65\%$ probability (from Appendix Figure C.1), the variance of the high- and low-frequency coefficient sets in each row is smaller than before the transformation. Please refer to the details in in the Appendix C.
>
> * The additional computational cost for inference is $\mathcal{O}(d)$ [2], where $d$ is the input length, as the Haar transform can be implemented using local convolutional layers, resulting in lower cost than methods such as FrameQuant.
>
> [1] Huang W, Liu Y, Qin H, et al. BiLLM: pushing the limit of post-training quantization for LLMs. Proceedings of the 41st ICML. 2024: 20023-20042.
>
> [2] Meyer, Y. and Ryan, R.D., Wavelets: Algorithms & Applications, Society for Industrial and Applied Mathematics, 1993.
>
> **Q1:** What is the difference between the proposed saliency-driven selection via L2 norm and AWQ?
>
> **Response:** The key difference lies in how saliency is defined and computed.
>
> AWQ selects salient weights based on the output of activation functions — it analyzes the activation-induced importance of weights during inference and identifies those with the greatest impact on model outputs. In contrast, our method in HBLLM performs analysis on the weight matrix itself: it computes the $\ell_2$ norm of each column in a saliency factor matrix derived directly from the weights, without relying on input-dependent activations.
>
> This makes HBLLM activation-free and efficient, as it avoids collecting runtime statistics, while still effectively identifying structurally significant components for preserving accuracy under quantization.
>
> We thank the reviewer for pointing out AWQ and will include a discussion comparing the two methods in the revised manuscript.
>
> **Q2:** What would happen if the reconstruction loss also considered the calibration data?
>
> **Response:** We tried incorporating calibration-data-based reconstruction loss. However, the results show no consistent improvements and sometimes even lead to degradation. The results are in the Table 1:
>
> **Table 1: Perplexity (↓, C4, Wiki2, PTB) and AvgQA accuracy (↑, AvgQA over 9 zero-shot tasks) of BiLLM variants with Haar. OPT-1.3B (cols 2–5), LLaMA2-7B (cols 6–9).**
>
> |Method|C4↓|Wiki2↓|PTB↓|AvgQA↑|C4↓|Wiki2↓|PTB↓|AvgQA↑|
> |-|-|-|-|-|-|-|-|-|
> |HBLLM-row|**19.55**|26.95|**25.70**|**46.87**|**13.00**|13.20|**85.50**|50.86|
> |HBLLM-col|21.98|**23.69**|27.39|45.21|13.18|**12.02**|146.1|51.34|
> |HBLLM-row+calib-loss|23.03|26.55|29.96|44.93|14.65|13.45|282.6|51.45|
> |HBLLM-col+calib-loss|24.95|28.47|36.89|44.27|14.19|13.41|247.3|**52.43**|
>
> **Q3:** Why does transforming weights to the frequency domain using the Haar Transform help capture the structural differences of weights?
>
> **Response:** While frequency transforms are commonly applied to features (e.g., images), our focus is the structure of weights. We assume different frequency bands encode different structural patterns. Applying Haar to weights enables selective grouping and quantization with minimal accuracy loss. This choice is motivated by two factors:
>
> * **Better grouping.** Frequency-domain representation increases the separability for grouping, enlarging the inverse quantization set (CIQ). As shown in Table 2, this improves fidelity over BiLLM. See Appendix B.3.
>
> * **Higher quantization efficiency.** Due to frequency sparsity, accurate weight recovery is possible using few components, reducing 1-bit quantization error. See Appendix C.3.
>
> **Table 2: Perplexity (↓, C4, Wiki2, PTB) and AvgQA accuracy (↑, AvgQA over 9 zero-shot tasks) of BiLLM variants with Haar. OPT-1.3B (cols 2–5), LLaMA2-7B (cols 6–9).**
>
> |Method|C4↓|Wiki2↓|PTB↓|AvgQA↑|C4↓|Wiki2↓|PTB↓|AvgQA↑|
> |-|-|-|-|-|-|-|-|-|
> |BiLLM|56.24|68.43|119.2|38.39|33.97|31.38|373.0|42.11|
> |BiLLM+$\ell_2^\dagger$|56.88|70.48|92.16|39.28|28.17|25.08|226.3|41.77|
> |HBLLM-row|**19.55**|26.95|**25.70**|**46.87**|**13.00**|13.20|**85.50**|50.86|
> |HBLLM-col|21.98|**23.69**|27.39|45.21|13.18|**12.02**|146.1|**51.34**|
>
> *Notes: $\ell_2$ denotes $\ell_2$-norm-based saliency-driven column selection; $\dagger$ denotes multi-parameter intra-row grouping. In Table 1, the results of HBLLM-row and HBLLM-col (our proposed method) already include these strategies.*
>
> **Q4:** In Table 1, could you provide the equation used to compute the average bit-width in detail? Since element-wise saliency detection creates overhead, why is the bit-width using HBLLM-col 1.00?
>
> **Response:** The average bit-width of a quantized matrix \\( \widehat{\mathbf{W}} \in \mathbb{R}^{n \times m} \\) is defined as the total memory cost \\( \mathcal{M} \\) divided by the number of elements:
>
> \\[
> \textbf{AvgBit} = \frac{\mathcal{M}}{n \times m}.
> \tag{1}
> \\]
>
> Let \\( \mathbf{W} \in \mathbb{R}^{n \times m} \\), \\( k \\) be block size , and \\( c \\) be the number of salient columns in \\( \mathbf{W} \\). For each method, $\mathcal{M}$ in eq. (1)  is defined by:
>
> \\[
> \mathcal{M}_{\text{BiLLM}} = 2nc + \left\lceil \frac{m}{k} \right\rceil \cdot 3n \cdot 16 + n(m - c) + \left\lceil \frac{m}{k} \right\rceil \cdot 2n \cdot 16 \cdot 2 + nm + \widetilde{m},
> \tag{2}
> \\]
>
> \\[
> \mathcal{M}_{\text{ARB-RC}} = 2nc + \left\lceil \frac{m}{k} \right\rceil \cdot (2n + 2c) \cdot 16 \cdot 2 + n(m - c) + \left\lceil \frac{m}{k} \right\rceil \cdot (n + m - c) \cdot 16 \cdot 2 + nm + \widetilde{m},
> \tag{3}
> \\]
>
> \\[
> \mathcal{M}_{\text{PBLLM}} = rnm + \left\lceil \frac{m}{k} \right\rceil \cdot 2n \cdot 16 + (1 - r)nm \cdot 8 + \left\lceil \frac{m}{k} \right\rceil \cdot 2n \cdot 16 + nm,
> \tag{4}
> \\]
>
> \\[
> \mathcal{M}_{\text{HBLLM-row}} = nm + \left\lceil \frac{m}{k} \right\rceil \cdot 3n \cdot 16 \cdot 2 + nc + \left\lceil \frac{m}{k} \right\rceil \cdot 2n \cdot 16 \cdot 2 + n(m + c) + \widetilde{m},
> \tag{5}
> \\]
>
> and
>
> \\[
> \mathcal{M}_{\text{HBLLM-col}} = n(m - c) + \left\lceil \frac{m}{k} \right\rceil \cdot 1.5n \cdot 16 \cdot 2 + nc + \left\lceil \frac{m}{k} \right\rceil \cdot 2n \cdot 16 \cdot 2 + nm + \widetilde{m},
> \tag{6}
> \\]
> where \\( r \\) denotes the ratio of the binarized weights.
>
> **Example: Average Bit-width of HBLLM**
>
> Assume \\( k = 128 \\), \\( c = 0.08m \\). For HBLLM-row and HBLLM-col, AvgBit will be:
>
> \\[
> \mathbf{AvgBit} = \frac{\mathcal{M}_{\text{HBLLM-row}}}{nm} = 2.16 + \frac{160n \left\lceil \frac{m}{k} \right\rceil}{nm} + \frac{\widetilde{m}}{nm} \approx 3.418~\text{bits},
> \tag{7}
> \\]
>
> and
>
> \\[
> \mathbf{AvgBit} = \frac{\mathcal{M}_{\text{HBLLM-col}}}{nm} = 2 + \frac{112n \left\lceil \frac{m}{k} \right\rceil}{nm} + \frac{\widetilde{m}}{nm} \approx 2.883~\text{bits},
> \tag{8}
> \\]
>
>  respectively.
>
> In HBLLM-col, saliency selection is performed at the column level rather than element-wise. Within each block, a few salient columns are identified via structured search and processed separately from the non-salient ones. Unlike HBLLM-row, HBLLM-col does not require FillAvg to complete the matrix, thus avoiding additional storage overhead.
>
> In summary, the only overhead in HBLLM-col is the **1-bit** binarization per weight matrix.
>
> Regarding the **equation numbering issue**, we will standardize the numbering of all equations in the final version.

---

> > ### Comment · Reviewer_cU7V · 2025-08-05
> >
> > Thank the authors for the response, this clarify most of my concern. AWQ points out the relationship between saliency in activations and in weights. In the paper, the authors use only weights and data-independence to identify salience, please correct me if I am wrong. Could the authors elaborate on and analyze the percentage of salient weights for each layer within a block and across different blocks?

---

> ### Author Response · Authors · 2025-08-06
>
> We appreciate the reviewer’s clarification and the opportunity to correct a misstatement in our previous response. In particular, our rebuttal may have caused confusion by stating that our method performs “analysis on the weight matrix itself.” This description was inaccurate.
>
> HBLLM captures the relationship between saliency in the Hessian matrix and in the weights. In practice, our method **does utilize input data** to compute the layer-wise Hessian matrix, which serves as part of the importance metric. Specifically, HBLLM estimates weight saliency based on $\mathbf{H} = 2\mathbf{X}\mathbf{X}^{\top}$ and the weight matrix $\mathbf{W}$, where $\mathbf{X}$ denotes calibration data. This computation appears in the line 1 of Algorithm 1 and is based on the formulation proposed in [1].
>
>
> **Regarding the reviewer's request for a quantitative analysis of salient weights across layers and blocks**: although we did not present such statistics in the original submission, we agree that this level of analysis provides valuable insight into how saliency is distributed throughout the network.
>
> To this end, we conducted two complementary analyses. First, we constructed a histogram to capture the saliency distribution across different types of weight layers and across transformer blocks. Here, saliency is defined as the ratio of salient (i.e., selected) weights to the total number of weights in the matrix. We divided the value range
> [0, 20%] into four equal-width bins and assigned each measurement to its corresponding bin. Table 1 summarizes the proportion of entries falling into each bin:
>
>
> **Table 1**. The histogram of the distribution of saliency  between different types of weight layers and across different transformer blocks.
>
> |bin|q|k|v|out|gate|up|down|transformer block|
> |-|-|-|-|-|-|-|-|-|
> |[0,    5%]|0.8125|0.8125|0.03125|0.15625|0.125|0|0.28125|0.03125|
> |(5%, 10%]|0.1875|0.1875|0.78125|0.59375|0.65625|0.5|0.71875|0.9375|
> |(10%, 15%]|0|0|0.1875|0.25|0.1875|0.375|0|0.03125|
> |(15%, 20%]|0|0|0|0|0.03125|0.125|0|0|
>
> Additionally, we present the statistical features of this data in Table 2.
>
> **Table 2**. The statistics of the distribution of saliency  between different types of weight layers and across different transformer blocks (in %).
>
> |layer type|q|k|v|out|gate|up|down|mean|transformer block|
> |-|-|-|-|-|-|-|-|-|-|
> |mean|4.32|4.15|8.28|7.52|7.67|10.62|5.40|6.85|7.30|
> |median|4.42|3.97|8.15|7.50|7.34|9.77|5.64|6.86|7.02|
> min|2.17|2.22|2.95|3.27|4.00|5.98|2.58|3.69|4.28|
> |max|5.76|6.35|13.01|12.23|15.01|17.65|6.81|9.83|10.72|
>
>
> Furthermore, we will illustrate the distribution of saliency in graphical form in the paper and analyze their clustering relationships.
>
>
> [1] Frantar, E., Singh, S. P., & Alistarh, D.  Optimal brain compression: a framework for accurate post-training quantization and pruning.
>   In Conference on Neural Information Processing Systems (NeurIPS), 2022.

---

> > ### Comment · Reviewer_cU7V · 2025-08-06
> >
> > Thank authors for the response. I am happy with this analysis. The analysis provided above could significantly strengthen the paper. Would it be possible for the authors to offer further insights based on Tables 1 and 2? Additionally, could you provide the distribution of saliency across different types of weight layers and transformer blocks, while also referencing saliency findings from previous works for comparison? Please feel free to disregard this request if it’s no longer relevant.

---

> ### Author Response · Authors · 2025-08-07
> **Further Insights into Layer-wise Saliency Distribution and Structural Patterns**
>
> Thank you for the follow-up and for highlighting the potential of the analysis to strengthen the paper. We would be happy to share further insights based on Tables 1 and 2. All the analyses in this discussion are based on experiments conducted with the LLaMA2-7B model.
>
> The saliency distribution analysis reveals several meaningful patterns. Notably, the query (**q**) and key (**k**) projection layers exhibit low saliency across the board, with over 80% of their weights falling into the [0%, 5%] bin. This suggests that these layers exhibit high compressibility and may contain a substantial proportion of redundant parameters. In contrast, the value (**v**), output (**out**), **gate**, and especially the **up** projection layers show significantly higher saliency, with a large fraction falling into the (5%, 10%] and (10%, 15%] bins. This indicates that these layers are structurally more important and require more careful treatment during quantization to preserve accuracy.
>
> From the statistical summary, we observe that the **up** projection layer has the highest mean saliency (10.62%), followed by the **v** and **gate** layers. This confirms that most of the critical information for model expressiveness tends to concentrate in these parts of the network. Conversely, the **q**, **k**, and **down** layers exhibit lower and more stable saliency, making them better candidates for more aggressive compression.
>
> Furthermore, we examined the *distribution of saliency* across different types of weight layers and transformer blocks, revealing how saliency evolves as the transformer blocks progress from input to output. Due to format constraints, we are unable to include the visualized data directly in this discussion response. However, the results show that the **q** and **k** layers share highly similar saliency distributions, while **gate** and **up** layers show comparable patterns. Likewise, the **v** and **out** projection layers exhibit similar saliency distributions. These relationships suggest an underlying structural alignment between specific layer types, which may be exploited for better-informed quantization strategies. We will include these extended analyses and discussion in the supplementary material of the revised submission.
>
> Regarding the suggestion to compare with prior work's saliency findings: we agree this would be a valuable addition. However, such a comparison would require additional experiments under consistent settings, which is beyond the scope of our current analysis. We plan to explore this direction in future work.
>
> Thank you again for the thoughtful suggestion and your positive feedback. We will integrate these insights and extended analyses into the revised version to improve the clarity and interpretability of our method.

---

> > ### Comment · Reviewer_cU7V · 2025-08-08
> >
> > Will the code be released soon? Thank authors for the detailed response. I think adding the above analysis to the revised paper will strengthen it. I will increase the score to 5 (accept).

---

> > > ### Author Response · Authors · 2025-08-08
> > >
> > > Thank you for your positive feedback and for increasing the score. We plan to release the code to support reproducibility and further research. In accordance with the double-blind review policy, the code will be made publicly available after the review process concludes, and the release link will be included in the final version of the paper.

---

### Official Review · Reviewer_VPLv · 2025-07-03

**Clarity:** 3
**Significance:** 4
**Originality:** 4
**Rating:** 6
**Confidence:** 5

**Summary:**

The submission proposes a new post-training LLM quantization technique based on frequency-domain lossy weight compression. It shares the main idea with other lossy frequency-domain compression schemes that were applied successfully to media data types in the past. Instead of exploiting different perception sensitivity levels of humans depending on frequency bands (works for media types), this method uses the Haar-transform, a saliency selection method of matrix columns and a new metric called CIQ that is used for the optimization.

**Questions:**

* Please add measurements showing inference speed (latency / throughput) especially as the technique requires a sequence of more complex math operations.

* Line 177: is K a single global parameter? Would multiple K work as well and if yes, how (maybe based on the energy of the Haar-spectrum)?

* Table 3 (FrameQuant), close to line 291: is it possible to get at least an estimate for the total amount of memory FrameQuant would have allocated?

**Ethical Concerns:**

["NO or VERY MINOR ethics concerns only"]

**Final Justification:**

All my issues were resolved by the rebuttal.

**Limitations:**

Yes

**Paper Formatting Concerns:**

Figure 2 maybe better placed closer to Section 3.2. The C(onference) / J(journal) suffixes with titles of the references are unusual and new to at least me. Either drop them (as most or even all are redundant) or add a legend to explain.

**Quality:**

4

**Strengths And Weaknesses:**

The strength of the submitted work is the idea. It seems to deliver consistently better performance at 1 bit-per-weight or close to that compared to >2 bits-per-weight of the next best performing technique which is FrameQuant.

Minor weaknesses are missing experimental results showing inference speed (latency / throughput) and a few details that are left unclear (see Questions).

---

> ### Author Rebuttal · Authors · 2025-07-31
>
> We sincerely appreciate the reviewer's positive acknowledgment of our work and their detailed suggestions. We would like to respond to the following
> three questions one by one.
>
> **Q1:** Add measurements showing inference speed (latency / throughput) especially as the technique requires a sequence of more complex math operations.
>
> **Response:** To address this point, we designed an experiment that combines direct measurement with estimation to evaluate the inference latency of HBLLM. This is necessary because, at present, there is no existing inference framework that fully supports the dequantization algorithm used in HBLLM. Building a highly optimized inference framework requires considerable engineering effort and cannot be completed in the short term.
>
> Our estimation results show that the inference latency of HBLLM is approximately $31.8$% of the FP16 baseline inference time.
>
> **Estimation Procedure**
>
> We define the relative inference time of HBLLM to the FP16 baseline as:
> \begin{equation}
>  R(p,l):= \frac{T_{HBLLM}}{T_{fp16}}=\left[ (1-p)+p/l \right].
> \tag{1}
> \end{equation}
> where:
> - $ T_{HBLLM} = p*T_{fp16} $ corresponds to the portion of inference time attributed to General Matrix–Vector multiplication (GEMV) ($p$ is the proportion of inference time spent on GEMV, which depends on the LLM architecture. Following GPTQ’s benchmark [1], we set $p=0.78$)
> - $l$ is the GEMV acceleration factor after quantization, defined as:
>  \begin{equation}
>  l:= \frac{T_{torch}}{ T_{hqmv}}.
> \tag{2}
> \end{equation}
> where $T_{torch}$ is the runtime of FP16 GEMV using PyTorch, and $T_{hqmv}$ is the runtime of our quantized GEMV implementation (denoted hqmv), which varies with the quantization strategy.
>
> Since the current hqmv implementation does not support Intra-Frequency Grouping (IFG), we use the runtime of HBLLM-col without IFG (denoted HBLLM-col w/o IFG) as a proxy to estimate the runtime of HBLLM-col with full IFG. Notably, applying IFG doubles the number of groups, potentially leading to:
> - roughly 2× more CUDA warp divergence,
> - a $90$% increase in average data loading (Avg bits per weight increases to 2.88).
>
> However, since data loading and computation can overlap on GPU, we conservatively estimate that the runtime with IFG should not exceed twice the runtime without IFG.
>
> **Experiment Steps**
>
> The full estimation process involves the following steps:
>
> 1. GEMV Runtime Measurement: Measure $T_{torch}$ and $T_{hqmv}$ using the HBLLM-col w/o IFG strategy.
>
> 2. Runtime Estimation with IFG: Estimate the runtime of hqmv with full HBLLM-col (i.e., with IFG), and compute the acceleration factor $l$.
>
> 3. Final Latency Ratio: Plug $p=0.78$ and the estimated $l$ into Equation (1) to compute the relative inference time
> $R$.
>
> **Experimental Results**
>
> Following the GPTQ benchmark setup [1], we tested GEMV on a single linear layer from the OPT-175B model. Experiments were run on an NVIDIA P100 GPU.
>
>
> The result from a single run showed $ T_{torch} = 1.35e-03s $ , and $ T_{hqmv} $ with HBLLM-col w/o IFG was $ 8.54e-05s $.
> Then, we estimated
> $T_{hqmv}$ with HBLLM-col w/o IFG to be $1.70e-04s$. Finally, substituting $p=0.78$ and $l=7.94$ into the equation yields
> $R=0.318.$
>
> This means that the estimated inference latency of HBLLM is only $31.8$% of that of the FP16 baseline, demonstrating its significant computational efficiency.
>
> [1] Elias Frantar, Saleh Ashkboos, Torsten Hoefler, Dan Alistarh.GPTQ: Accurate Post-training Compression for Generative Pretrained Transformers. arXiv preprint arXiv:2210.1732, 2022.
>
> **Q2:** Line 177: is K a single global parameter? Would multiple K work as well and if yes, how (maybe based on the energy of the Haar-spectrum)?
>
> **Response:** In HBLLM, the significant column count K is not a globally fixed hyperparameter; instead, it is adaptively determined as the optimal value within each matrix block. Therefore, there are multiple values of K.
>
> Specifically, the process for selecting significant columns is illustrated in  Appendix Algorithm E.1.
>
> For each block, we sort the columns based on their saliency scores in descending order and gradually accumulate the candidate significant columns, calculating their corresponding binary reconstruction errors. Ultimately, we select the number of significant columns n* that results in the minimum error as the optimal K for that block. In Algorithm E.1, n* should be changed to K to maintain consistency with the main text and eliminate any misunderstanding.
>
> Regarding the suggestion to "select salient columns based on the energy of the Haar spectrum," we find this to be very valuable. The saliency-driven algorithm presented in this paper does not currently consider this direction. In the future, we will explore incorporating Haar energy spectrum features into the criteria for selecting significant columns.
>
> **Q3:** Table 3 (FrameQuant), close to line 291: is it possible to get at least an estimate for the total amount of memory FrameQuant would have allocated?
>
> **Response:** Yes, the memory usage of FrameQuant can be estimated. We provide experimental results for smaller models and an extrapolated estimate for the larger one.
>
> Specifically, we conducted experiments to measure the actual GPU memory consumption during the quantization process of LLaMA1-7B and LLaMA1-13B using FrameQuant. The results are shown in Table 1.
>
> **Table 1: GPU Memory Usage(GB) during FrameQuant Quantization**
>
> |Model|Memory(GB)|
> |-|-|
> |LLaMA1-7B|11|
> |LLaMA1-13B|16|
>
> Based on these experimental results, we estimate the memory usage for LLaMA1-30B to be approximately **40 GB**. During the quantization of the 30B model, this estimated requirement exceeded available GPU memory, leading to runtime failures such as the one mentioned near line 291 in the main text.
>
>
> **Paper Formatting Concerns:** Additionally, regarding the layout position and citation format of Figure 2, we appreciate the reviewer’s detailed suggestions. We will adjust the position of Figure 2 in the final version, moving it closer to Section 3.2 to enhance the readability of the corresponding process description.
>
> As for the "C" / "J" designations in the references, they represent conference and journal sources, respectively, and are primarily used for distinguishing between document types. In the revised version, we will remove these suffixes to improve formatting consistency and avoid confusion.

---

> > ### Comment · Reviewer_VPLv · 2025-08-06
> >
> > Thank you for your detailed answers. I suggest that you add your answers to the submission in a reasonable form and the additional numbers of Table 1 in your rebuttal to Table 4 in the submission.

---

> > > ### Author Response · Authors · 2025-08-07
> > >
> > > Thank you again for your helpful comments and suggestions. We appreciate your acknowledgment and are glad that the responses were helpful. As suggested, we will incorporate the additional measurements and clarifications from the rebuttal into the final version of the submission, including integrating the extra numbers from Table 1 into Table 4. We believe these additions will strengthen the clarity and completeness of the paper.

---

> ### Author Response · Authors · 2025-08-06
> **Official Comment by Authors**
>
> Dear Reviewer VPLv,
>
> We appreciate your constructive comments and have provided detailed responses to your concerns. We would like to invite you to discuss any points that may still need clarification or further attention. Your feedback is invaluable to us, and we look forward to your insights.
>
> Thank you!
> Authors

---

### Official Review · Reviewer_3GV9 · 2025-07-03

**Clarity:** 2
**Significance:** 2
**Originality:** 2
**Rating:** 4
**Confidence:** 4

**Summary:**

This paper proposes HBLLM, a 1 bit PTQ method for LLM. The key idea is to apply a Haar wavelet transform to decompose weight matrices into frequency components, enabling structure-aware quantization. The method further introduces two grouping strategies: frequency-aware intra-row grouping and saliency-driven column selection, alongside a shared-mean quantization scheme for non-salient components. Experiments show improved perplexity and QA performance compared to prior 1-bit quantization methods.

**Questions:**

(1) I encourage the authors to disentangle the contribution of Haar-based frequency decomposition from other components (e.g., grouping, saliency selection) to provide a clearer understanding of where the benefits originate.

**Ethical Concerns:**

["NO or VERY MINOR ethics concerns only"]

**Final Justification:**

The authors' response addressed my concern. Thus, I decided to raise my rating of this paper.

**Limitations:**

The authors did not discuss the limitations of the work.

**Paper Formatting Concerns:**

See comments above.

**Quality:**

3

**Strengths And Weaknesses:**

Strengths

(1) Introduces the Haar wavelet transforms as a localized orthogonal transformation to increase binary expressiveness.
(2) Introduces two innovative grouping strategies, frequency-aware multi-parameter intra-row grouping and l2-norm saliency-driven column selection, and also employs a shared mean for non-salient weights, to better preserve important weight structures.

Weaknesses

(1) While the Haar transform is well-motivated theoretically, its contribution is not isolated. The authors do not compare their method with a baseline where grouping and saliency selection are applied directly in the original weight space without the Haar transform. This leaves unclear whether the Haar decomposition meaningfully improves quantization results beyond what grouping and saliency alone achieve.

(2) Fig.2 is not clear enough, maybe adding more description in the caption would help understanding. Also, if could add a figure to illustrate the weight decomposetion (before and after) would greatly enhance the reader's understanding of the core idea.

---

> ### Author Rebuttal · Authors · 2025-07-31
>
> We sincerely thank the reviewer for the thorough evaluation and insightful suggestions. We fully agree that verifying the individual contribution of each component and improving the clarity of our results are both important and necessary.
>
> **Q1 (W1):** I encourage the authors to disentangle the contribution of Haar-based frequency decomposition from other components (e.g., grouping, saliency selection) to provide a clearer understanding of where the benefits originate.
>
> **Response:** To address this, we conducted additional experiments to explicitly isolate and validate the performance gain brought by the Haar transform in our method.
>
> The results confirm that the Haar transform indeed lead to significant improvements in HBLLM performance — gains that cannot be achieved by other strategies such as grouping or saliency selection alone.
>
> Specifically, starting from the original BiLLM baseline, we independently activated the following two strategy components:
>
> - L2-norm-based saliency-driven column selection,
>
> - Frequency-aware multi-parameter intra-row grouping.
>
> We then compared their performance with HBLLM variants that incorporate the Haar transform, with or without these two additional strategies.
>
> As shown in the Table 1, although introducing either the $\ell_2$-based saliency or the aware grouping alone brings minor improvements to BiLLM, the addition of the Haar transform in HBLLM consistently yields more substantial gains in perplexity reduction and QA accuracy improvement on both OPT-1.3B and LLaMA2-7B, even under partial strategy activation (e.g., HBLLM-col+$\ell_2$).
>
> **Table 1: Perplexity (↓, C4, Wiki2, PTB) and AvgQA accuracy (↑, AvgQA over 9 zero-shot tasks) of BiLLM variants with Haar. OPT-1.3B (cols 2–5), LLaMA2-7B (cols 6–9).**
>
> |Method|C4↓|Wiki2↓|PTB↓|AvgQA↑|C4↓|Wiki2↓|PTB↓|AvgQA↑|
> |-|-|-|-|-|-|-|-|-|
> |BiLLM|56.24|68.43|119.2|38.39|33.97|31.38|373.0|42.11|
> |BiLLM+$\ell_2$|55.95|72.42|105.9|37.95|33.46|31.34|695.8|41.11|
> |BiLLM+$\ell_2^\dagger$|56.88|70.48|92.16|39.28|28.17|25.08|226.3|41.77|
> |Row-Haar+BiLLM|47.45|52.81|62.81|39.57|25.77|25.12|138.0|44.67|
> |Col-Haar+BiLLM|95.56|128.8|171.3|36.92|41.03|37.25|5193|39.60|
> |HBLLM-row+$\ell_2$|26.47|33.68|41.17|41.17|16.26|19.86|87.90|47.90|
> |HBLLM-col+$\ell_2$|26.37|29.99|36.24|42.13|15.04|13.99|154.6|49.38|
> |HBLLM-row|**19.55**|26.95|**25.70**|**46.87**|**13.00**|13.20|**85.50**|50.86|
> |HBLLM-col|21.98|**23.69**|27.39|45.21|13.18|**12.02**|146.1|**51.34**|
>
> *Notes: $\ell_2$ denotes $\ell_2$-norm-based saliency-driven column selection; $\dagger$ denotes multi-parameter intra-row grouping. In Table 1, the results of HBLLM-row and HBLLM-col (our proposed method) already include these strategies.*
>
> In summary, the Haar transform itself plays a decisive role in improving the final performance. We will include the corresponding experiments and analysis in the final version of the paper.
>
> **W2:** Fig.2 is not clear enough, maybe adding more description in the caption would help understanding. Also, if could add a figure to illustrate the weight decomposetion (before and after) would greatly enhance the reader's understanding of the core idea.
>
> **Response:** We appreciate the reviewer’s feedback about the lack of clarity in Figure 2. Due to rebuttal constraints, we cannot update the figure at this stage, but we provide the following clarification in text.
>
> The HBLLM quantization process consists of four steps: preparation, salient column selection, haar transform and quantization for the salient part, and quantization for the non-salient part.
>
> Since the salient columns are excluded from the Haar transform of the non-salient part, their positions must be filled before performing row-wise Haar transforms. This is handled by a process we refer to as FillAvg, where each missing column is filled with the average of its adjacent non-salient columns.
>
> For the non-salient part, HBLLM supports flexible choice between row-wise (HBLLM-row) and column-wise (HBLLM-col) transforms. The salient part undergoes column-wise Haar transformation followed by HaarQuant for quantization.
>
> We will further revise the caption in the final version.
>
> In the final version, we will also include additional visualizations (e.g., energy distribution before and after Haar transform, examples of row/column grouping structures) to aid understanding.

---

> ### Author Response · Authors · 2025-08-06
> **Official Comment by Authors**
>
> Dear Reviewer 3GV9,
>
> We appreciate your constructive comments and have provided detailed responses to your concerns. Should there be any aspects that require further clarification or discussion, we would be grateful for your feedback.
>
> Your insights are invaluable to us, and we look forward to hearing from you.
>
> Thank you!
>
> Authors

---

> > ### Comment · Reviewer_3GV9 · 2025-08-06
> >
> > I have checked the author's response. They have addressed my concerns properly.

---

> > > ### Author Response · Authors · 2025-08-07
> > >
> > > Thank you for your review and for acknowledging our response. We appreciate your earlier suggestions regarding the need to disentangle the Haar-based decomposition from other components and to improve the visual clarity of Fig. 2. We will revise the manuscript accordingly by including an ablation study that isolates the contribution of the Haar transform and by enhancing the figure and caption to better illustrate the weight decomposition process. We believe these revisions will improve the clarity and transparency of our contributions.

---

### Decision · Program_Chairs · 2025-09-17

**Decision:**

Accept (spotlight)

**Comment:**

This paper proposes HBLLM, a wavelet-enhanced high-fidelity 1-bit post-training quantization method for Large Language Models that leverages Haar wavelet transforms and innovative structure-aware grouping strategies to improve quantization fidelity with minimal overhead. All the reviewers posted positive comments After discussion, most of the concerns have been well resolved. The method demonstrates significant strengths, including the novel integration of Haar wavelet transforms to enhance binary expressiveness, innovative structure-aware grouping strategies that effectively preserve important weight structures. Therefore, the AC recommends to accept this paper.